# Bridging Scales: Spectral Theory Reveals How Local Connectivity Rules Sculpt Global Neural Dynamics in Spatially Extended Networks

**Yuhan Huang**[1,2]    **Keren Gao**[1]    **Dongping Yang**[3]    **Sen Song**[4]    **Guozhang Chen**[1]*

[1] School of Computer Science, Peking University, China
[2] School of Information Science and Technology, University of Tokyo, Japan
[3] Zhejiang Lab, China
[4] Department of Biomedical Engineering, Tsinghua University, China
`guozhang.chen@pku.edu.cn`

## Abstract

The brain's diverse spatiotemporal activity patterns are fundamental to cognition and consciousness, yet how these macroscopic dynamics emerge from microscopic neural circuitry remains a critical challenge. We take a step in this direction by developing a spatially extended neural network model integrated with a spectral theory of its connectivity matrix. Our theory quantitatively demonstrates how local structural parameters, such as E/I neuron projection ranges, connection strengths, and density determine distinct features of the eigenvalue spectrum, specifically outlier eigenvalues and a bulk disk. These spectral signatures, in turn, precisely predict the network's emergent global dynamical regime, encompassing asynchronous states, synchronous states, oscillations, localized activity bumps, traveling waves, and chaos. Motivated by observations of shifting cortical dynamics in mice across arousal states, our framework not only provides a possible explanation for repertoire of behaviors but also offers a principled starting point for inferring underlying effective connectivity changes from macroscopic brain activity. By mechanistically linking neural structure to dynamics, this work advances a principled framework for dissecting how large-scale activity patterns—central to cognition and open questions in consciousness research—arise from, and constrain, local circuitry. The implementation code is available at `https://github.com/huang-yh20/spatial-linear-project`.

## 1   Introduction

The brain's activity is remarkably diverse, forming complex spatiotemporal patterns that vary with an organism's cognitive state and level of consciousness [1–4]. Propagating waves of neural activity, for example, are observed across numerous brain regions and species, from insects to humans [5, 6]. Recent high-resolution imaging in mice, as they transition from anesthesia to wakefulness, has highlighted this complexity, revealing large-scale cortical waves during anesthesia and a shift towards more localized, intricate spatiotemporal patterns upon awakening [4]. Such observations emphasize that the spatial organization of neural activity is a fundamental aspect of brain function [7–10]. Consequently, a key challenge is to understand how these macroscopic dynamical states, which distinguish brain states, arise from the underlying microscopic neural architecture and effective connectivity rules. Since many critical brain dynamics are inherently spatial, involving coordinated

---

*Corresponding author: `guozhang.chen@pku.edu.cn`

39th Conference on Neural Information Processing Systems (NeurIPS 2025).

activity across neural tissue, models must incorporate spatial extent to capture these phenomena faithfully [11].

Previous developments in neural field theory have revealed a variety of rich dynamical phenomena in spatially distributed neural networks, such as traveling waves and bump-like activity patterns [12–18]. Meanwhile, previous development of dynamical mean-field theory (DMFT) [19–25] has uncovered the emergence of chaotic neural activity in large-scale brain networks. However, A comprehensive framework that quantitatively links a broad range of local network structures to the full spectrum of emergent global dynamics in spatially extended systems is still developing. Current theories often explain particular aspects, but a general theory predicting the emergence of, and transitions between, diverse spatiotemporal patterns from fundamental structural properties across a wide parameter space remains a critical need. For a detailed comparison with prior approaches, including neural field theory, numerical simulation and random matrix theory, please refer to Appendix A.1.

To address this, we present a unifying theoretical framework centered on a spatially extended recurrent neural network model with excitatory (E) and inhibitory (I) populations. The core of our approach is a spectral analysis of the network's connectivity matrix. Building upon theories of random matrices [26–30], we provide an analytical formulation for the spectral bulk arising from connection heterogeneity and, critically, demonstrate that the inherent spatial organization of the network itself constitutes a low-rank structure. This allows us to characterize a rich set of outlier eigenvalues, reflecting specific spatiotemporal modes determined by the network's geometry. We show that this comprehensive "spectral blueprint" - encompassing both the derived bulk and the spatially determined outliers - quantitatively links key local structural parameters (spatial reach of E/I connections, relative strengths, local density, weight variability) to a full repertoire of global dynamical phases. This framework offers predictive power, enabling us to anticipate the network's spatiotemporal behavior from its effective connectivity structure. These predicted phases include stable asynchronous states, global synchrony, oscillations, localized bumps, traveling waves, and chaotic dynamics. Our work thus provides a principled understanding of how network structure dictates emergent neural activity.

**Road-map** We first define the spatial E/I network (Section 2), then analytically derive its bulk-plus-outlier spectrum (Section 3.2, Appendix A.13). The spectrum predicts six dynamical phases (Section 3.3, Appendix A.11 & A.14). The comparison to experimental data can be found in Section 3.5-3.6. Parameters and numerical validation are in Appendix A.5-A.7, A.10.

## 2 Spatial Extended Neural Networks

### 2.1 Model Definition

To explore the link between structure, dynamics, and the eigenvalue spectrum, we constructed a minimal rate-based neural network model with biologically realistic features. The network consists of excitatory and inhibitory neurons, sparse connectivity that decays with distance, and synaptic weights drawn from a Gaussian distribution (Fig. 1).

We consider a network consisting of $N_E$ excitatory neurons and $N_I$ inhibitory neurons, where $N_E : N_I = 4 : 1$. Both excitatory neurons and inhibitory neurons are evenly distributed in the region $[0, 1) \times [0, 1)$. The dynamics of the neurons are described by the following equation:

$$\tau \frac{dh_i^\alpha}{dt} = -h_i^\alpha + \sum_\beta \sum_j J_{ij}^{\alpha\beta} \phi_\beta(h_j^\beta) + \xi_i^\alpha(t). \tag{1}$$

Here, $h$ indicates the membrane potential of neurons, $\phi(\cdot)$ is the activation function of neurons, $\alpha, \beta \in \{E, I\}$ denotes the neuronal populations, $\xi^\alpha$ represents the external input received by the population $\alpha$ , and $\mathbf{J}$ represents the synaptic weights between neurons. The external input is modeled as independent white noise with intensity $\xi_0 = 0.1$.

The connections between neurons are sparse. The probability of connection between a $\alpha$ neuron located at $x_i$ and a $\beta$ neuron located at $x_j$ is given by,

$$p_c^{\alpha\beta}(x_i - x_j; y_i - y_j) = \frac{k_{\alpha\beta}^{out}}{N_\alpha} g(x_i - x_j; d_{\alpha\beta}) g(y_i - y_j; d_{\alpha\beta}), \tag{2}$$

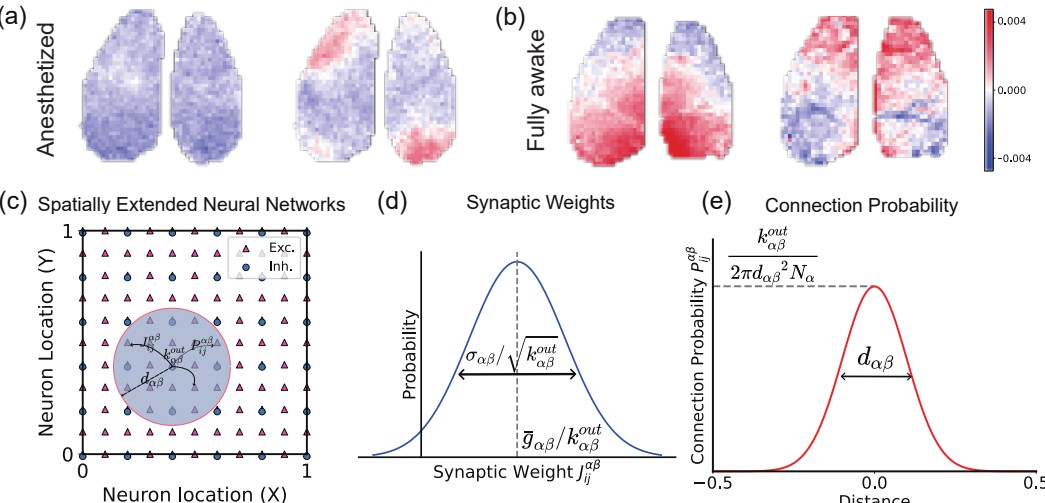

Figure 1: (a) and (b) Mesoscopic optical imaging of the anesthetized and full awake mouse cortex [4]. Model schematic. (c) Spatially extended neural network with excitatory (E) and inhibitory (I) neurons embedded in 2D space. (d) Synaptic weights follow a distance-dependent Gaussian distribution. (e) Connection probability follow a distance-dependent wrapped Gaussian distribution.

where

$$g(x_i - x_j; d_{\alpha\beta}) = \frac{1}{\sqrt{2\pi}d_{\alpha\beta}} \sum_{n=-\infty}^{\infty} e^{-\frac{(x_i - x_j + n)^2}{2d_{\alpha\beta}^2}} .$$

This indicates that the probability of connection between neurons decays according to a wrapped Gaussian profile with a characteristic decay length of $d_{\alpha\beta}$. Here, $k_{\alpha\beta}^{out}$ represents the average out-degree, namely the number of $\alpha$ neurons to which a single $\beta$ neuron projects.

Two neurons are connected with a probability of $p_c^{\alpha\beta}(|x_i - x_j|)$. If two neurons are indeed connected, the connection weight follows an independent Gaussian distribution with a mean of $\bar{g}_{\alpha\beta}/k_{\alpha\beta}^{out}$ and a variance of $\sigma_{\alpha\beta}^2/k_{\alpha\beta}^{out}$, namely, $J_{ij}^{\alpha\beta} \overset{\text{i.i.d.}}{\sim} \mathcal{N}(\bar{g}_{\alpha\beta}/k_{\alpha\beta}^{out}; \sigma_{\alpha\beta}^2/k_{\alpha\beta}^{out})$.

The activation function $\phi(\cdot)$ can be arbitrary. In our experiment, we chose the activation function $\phi_E(x) = \tanh(x)$ for excitatory neurons and $\phi_I(x) = 5\tanh(x/5)$ for inhibitory neurons for the reason that the saturated firing rate of inhibitory neurons is substantially higher than that of excitatory neurons[31]. However, the choice of activation function has little impact on the conclusions, and we discuss this in detail in the Appendix A.6. In the Appendix A.6, we also show the results with the activation function as $\phi_E(x) = 10 \cdot ReLU(x)$ and $\phi_I(x) = 2 \cdot ReLU(x)^2$.

## 2.2 A Repertoire of Emergent Dynamical Phases

**Asynchronous State** The balance between excitatory and inhibitory interactions shapes the network's synchronization and stability. When inhibition dominates or excitation and inhibition are balanced (Fig. 2(a)), the network exhibits low synchronization and small deviations from the steady firing rate. We define this regime as the asynchronous state (Fig. 2(c)). In the asynchronous state, neuronal membrane potentials fluctuate around the fixed point (zero) due to external input. Neural activity remains weakly correlated across neurons, and no spatial patterns emerge. This regime aligns with the classical asynchronous state described by [21].

**Synchronous State** When excitation dominates (Fig. 2(a)), the network can enter a highly synchronized regime with large deviations from the steady-state firing rate, which we term the synchronous state phase. In this phase, the firing rates of all neurons shift away from the fixed point, and strong global synchronization emerges across the network (Fig. 2(c)). This globally synchronized activity resembles pathological brain states such as epileptic seizures [32].

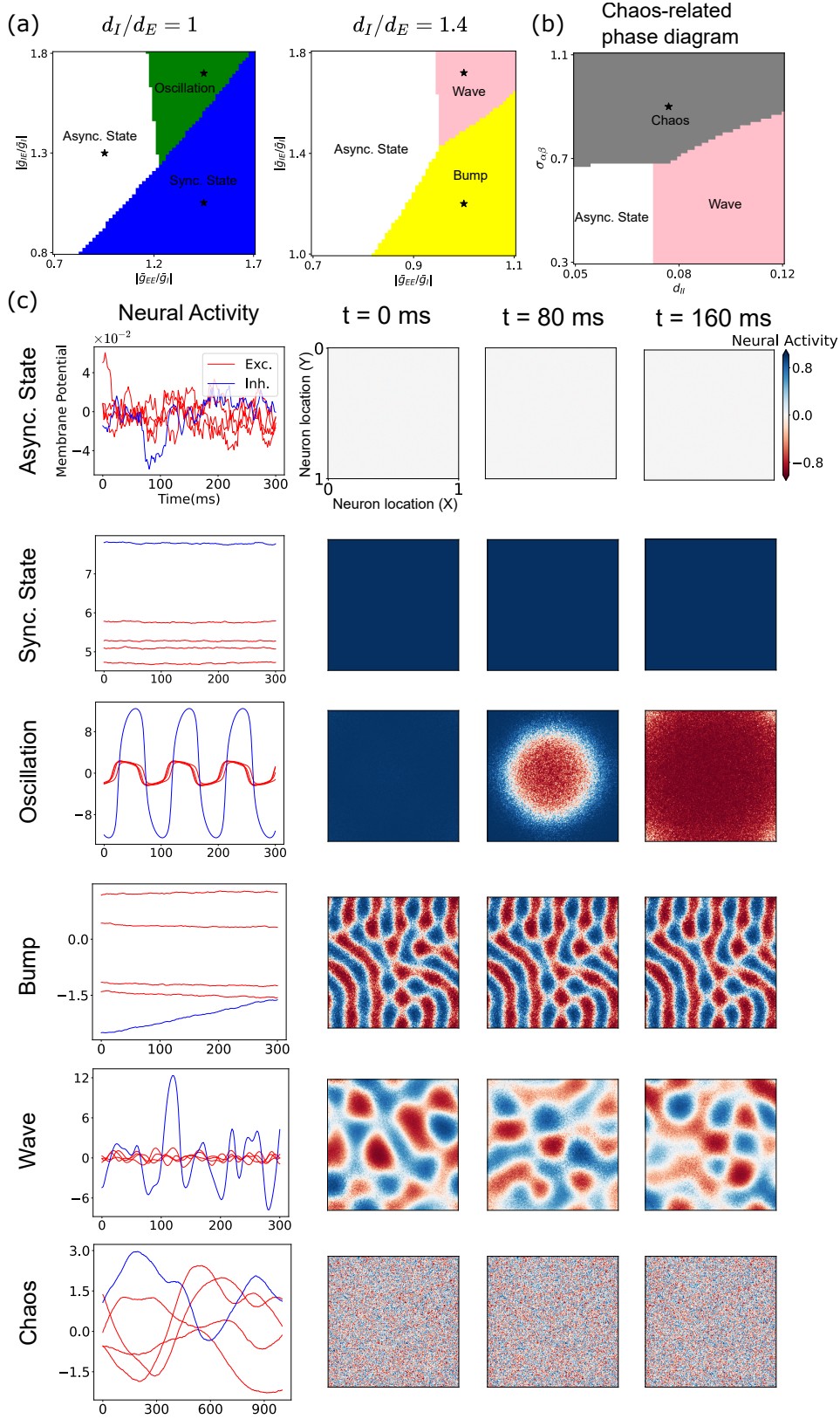

Figure 2: Dynamical regimes in spatially extended networks. (a,b) Phase diagrams under different parameters (In simulation we set $\bar{g}_{EI} = \bar{g}_{II} = \bar{g}_I$, $\sigma_{EE} = \sigma_{IE} = \sigma_{EI} = \sigma_{II} = \sigma_{\alpha\beta}$, see Appendix A.10). Wave-chaos phase boundaries determined as in Appendix A.14. (c) Representative neural activity showing temporal evolution (left) and spatial patterns (right) with shared colorbar; enhanced versions in Appendix A.12 show the asynchronous state with individual scaling.

**Oscillatory Phase**    Neural oscillations emerge when excitation-inhibition coupling strengthens in highly synchronized networks (Fig. 2(a)). We term this dynamical regime the oscillatory phase, characterized by periodic membrane potential changes and synchronized neural activity (Fig. 2(c)). Our identified excitation-inhibition loop mechanism aligns with established experimental and theoretical work. This consistency supports the biological plausibility of our model, as evidenced by prior studies on neural oscillations [1, 33].

**Localized Bump Phase**    Mismatched projection ranges between neuron types generate structured spatial activity patterns. When excitatory neurons project locally while inhibitory neurons extend farther, the system exhibits bump phases and wave phases (Fig. 2(a)). The bump phase produces localized spatial patterns dependent on activation functions. Neural activity synchronizes within discrete regions, forming stripes (Fig. 2(c)). Rectified linear and threshold power-law functions yield spot-like patterns (see Appendix A.6), aligning with previous simulation results of spiking neural networks[15, 17]. Similar wavelength-specific patterns occur in juvenile visual cortex [34], with persistent long-range correlations in adulthood [35].

**Wave Phase**    The wave phase features propagating oscillations with location-dependent phase shifts (Fig. 2(c)). This mirrors biological observations of traveling alpha/gamma waves [6, 5], suggesting our model captures essential mechanisms of spatial-temporal dynamics.

**Chaotic Phase**    High connection sparsity and weight variance drive neural activity into a chaotic regime (Fig. 2(b) and Fig. 9(a)). We term this disordered state the chaos phase, where spatial patterns collapse due to uncorrelated neural firing. Neurons exhibit large-amplitude, weakly correlated fluctuations in the chaos phase (Fig. 2(c)). Membrane potentials vary erratically, matching the second type of asynchronous state described in spiking networks [21]. This aligns with dynamic mean-field theory predictions of chaotic dynamics in rate-based neural networks [19].

## 3    The Spectral Blueprint: Decoding Dynamics from Connectivity Structure

### 3.1    Effective Connectivity and its Eigenvalues

Building on the neuroscience perspective of effective connectivity (EC) as a model-based measure of directed interactions [36], we propose that the linearized Jacobian matrix $-\mathbf{I} + \mathbf{J}\phi'(\mathbf{x}^*)$–derived from the recurrent neural network's fixed-point dynamics–serves as an analogous EC matrix. Here, structural connectivity $\mathbf{J}$ is dynamically modulated by nonlinear gains $\phi'(\mathbf{x}^*)$, mirroring how anatomical constraints and state-dependent plasticity jointly shape brain network interactions. To unravel how such connectivity shapes collective behavior, we analyze its eigenvalue spectrum, which governs stability and activity patterns. In the following sections, we employ random matrix theory to characterize this spectrum, revealing universal dynamical regimes emergent from its structure. In the main text, we use the tanh activation function, so the connectivity matrix $\mathbf{J}$ can be regarded as the effective interaction matrix. We also present results with alternative activation functions in the appendix A.6 and A.11.

### 3.2    Eigenvalue Spectrum of Spatially Extended Networks

The eigenvalue spectrum of spatially distributed neural networks' connectivity matrix $\mathbf{J}$ comprises two distinct components: a bulk disk region and a set of spectral outliers, which respectively reflect heterogeneous neuron interactions and population-averaged connectivity patterns. Following the matrix decomposition $\mathbf{J} = \bar{\mathbf{J}} + \delta\mathbf{J}$, where $\bar{\mathbf{J}}$ denotes the expectation matrix and $\delta\mathbf{J}$ represents zero-mean random fluctuations, we observe that $\delta\mathbf{J}$ governs the bulk spectrum through its stochastic components while $\bar{\mathbf{J}}$ generates spectral outliers through its low-rank structure. This aligns with the perturbation framework established by [27], wherein low-rank modifications to random matrices predominantly affect outlier positioning while preserving the original bulk spectral radius (see Appendix A.13.2 for details). The dichotomy between these spectral components provides a mathematical characterization of neural network dynamics - the deterministic outliers capture macroscopic interaction features, whereas the bulk spectrum encodes microscopic connection variability.

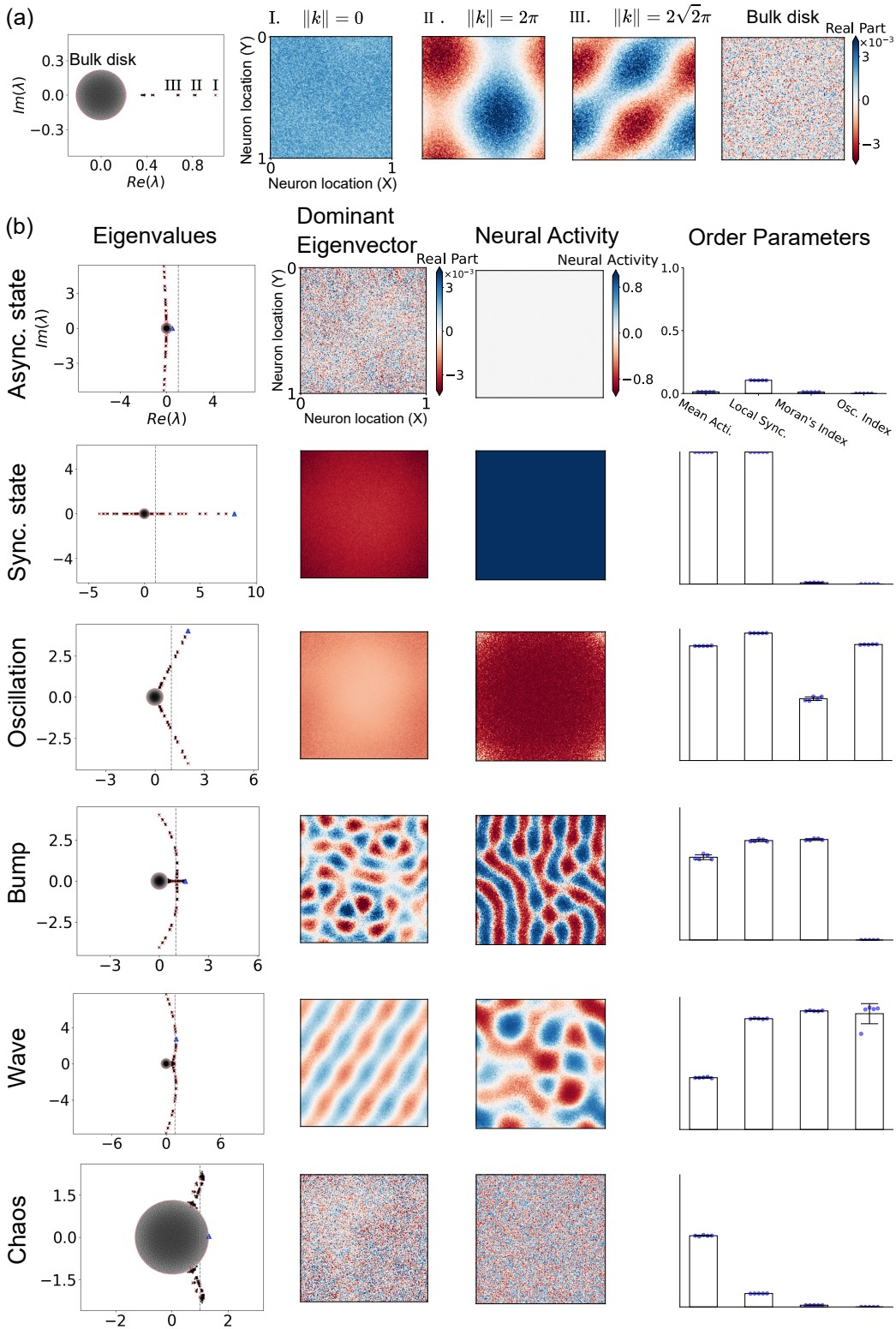

Figure 3: Spectral signatures and spatial patterns of network dynamics. (a) Eigenvalue spectra and corresponding eigenvectors across spatial modes. (b) Example eigenvalue distributions (blue triangles: eigenvalues with the largest real part), leading eigenvectors, neural activity, and order parameters (blue point: data points) for distinct dynamical states. The four order parameters (rightmost column) indicate, from left to right: neural activity fluctuations, local synchrony, spatial patterning, and oscillations. Detailed definitions are given in the Appendix A.4.

### 3.2.1 Outliers

The matrix $\bar{\mathbf{J}}$ determines the outlier part of the eigenvalue spectrum [27]. The expected connectivity between neurons exhibits spatial translational invariance. We can decompose the activities of neurons into different spatial Fourier modes (See Appendix A.13.4). For each spatial Fourier mode, the effective connectivity matrix is

$$
\overline{J}_{\text{eff}}^{(k)} = \begin{bmatrix} \overline{g}_{EE} \exp\left(-\frac{||k||^2 d_{EE}^2}{2}\right) & \overline{g}_{IE} \exp\left(-\frac{||k||^2 d_{IE}^2}{2}\right) \\ \overline{g}_{EI} \exp\left(-\frac{||k||^2 d_{EI}^2}{2}\right) & \overline{g}_{II} \exp\left(-\frac{||k||^2 d_{II}^2}{2}\right) \end{bmatrix},
\tag{3}
$$

where $\vec{k} = (2\pi n_x, 2\pi n_y)$, $\quad n_x, n_y = 0, \pm 1, \pm 2, \cdots$.

The eigenvalue spectrum of the connectivity matrix reveals distinct outliers generated by spatial Fourier modes at different wave vectors $k$. Each outlier corresponds to collective population dynamics mediated by $k$-specific interaction submatrices. These submatrices encode spatially modulated couplings between excitatory and inhibitory populations, with effective weights determined by three components: (1) baseline inter-population connectivity ($\overline{g}_{\alpha\beta}$ between populations $\alpha$ and $\beta$), (2) $k$-dependent modulation of spatial interaction range $\exp\left(||k||^2 d_{\alpha\beta}^2/2\right)$.

### 3.2.2 Bulk Disk

The matrix $\delta\mathbf{J}$ governs the bulk disk part of the eigenvalue spectrum, of which the radius is determined by the local sparsity of neuronal connections and the variance of the weights. Analytical tools developed in [28, 29] characterize the spectral distribution of random matrices with independent, zero-mean, finite-variance entries. Applying these results, we can determine that the eigenvalues are extended within a circle of a specific radius $r = \sqrt{\max(\lambda(M))}$, where the elements of the matrix $M$ represent the variances of the elements of the connectivity matrix $\mathbf{J}$. The variance of the elements of the connectivity matrix $\mathbf{J}$ is given by (See Appendix A.13.5 for details),

$$
M_{ij}^{\alpha\beta} = \mathbb{E}\left[\delta J_{ij}^{\alpha\beta 2}\right] = p_c^{\alpha\beta}\left(|x_i - x_j|\right)\left(1 - p_c^{\alpha\beta}\left(|x_i - x_j|\right)\right) \times \frac{\overline{g}_{\alpha\beta}^2}{k_{\alpha\beta}^{out^2}} + p_c^{\alpha\beta}\left(|x_i - x_j|\right)\frac{\sigma_{\alpha\beta}^2}{k_{\alpha\beta}^{out}}.
\tag{4}
$$

The eigenvalues of this matrix are equivalent to those of the reduced matrix, where the elements represent the heterogeneity of connections between different types of neurons. For the two-dimensional case, this reduced matrix is

$$
M_{\alpha\beta} = \frac{N_\beta}{N_\alpha}\left[\frac{\overline{g}_{\alpha\beta}^2}{k_{\alpha\beta}^{out}}\left(1 - \frac{k_{\alpha\beta}^{out}}{4\pi d_{\alpha\beta}^2 \cdot N_\alpha}\right) + \sigma_{\alpha\beta}^2\right].
\tag{5}
$$

Based on the above equations, the radius of the circular part of the eigenvalue spectrum is given by,

$$
r = \sqrt{\frac{M_{EE} + M_{II} + \sqrt{(M_{EE} - M_{II})^2 + 4M_{EI}M_{IE}}}{2}}.
\tag{6}
$$

We can observe that the heterogeneous matrix $M_{\alpha\beta}$ is composed of two parts: sparsity and variability in synaptic weights. The left half of Equation 5 represents the heterogeneity brought about by the sparse connections between neurons, while the right half of Equation 5 represents the heterogeneity due to the variability in the weights of the neuronal connections. As the local sparsity of neuronal connections and the variance of weights increase, the radius of the bulk disk part of the eigenvalue spectrum also increases.

### 3.3 Linking Spectral Features to Dynamical Phases

The eigenvalue spectrum of the effective connectivity matrix $\mathbf{J}\phi'(x^*)$ governs network dynamics: the dominant eigenvalue (largest real part) determines stability near the fixed point. A real part exceeding the critical threshold ($Re(\lambda_{\text{dom}}) \geq 1$) quantifies deviation magnitude, while the dominant eigenvector specifies the spatial activity pattern. We thus classify neural networks into distinct *dynamical phases* based on their eigenvalue spectrum, as summarized in the table 1. (See Appendix A.11 for details),

Table 1: Dynamical Phases and Spectral Features

| Phase | $\Re(\lambda_{\mathbf{dom}})$ **Condition** | $\lambda_{\mathbf{dom}}$ **Type** | **Wavenumber (k)** |
|---|---|---|---|
| Asynchronous State | $\Re(\lambda_{\text{dom}}) < 1$ | - | - |
| Synchronized State | $\Re(\lambda_{\text{dom}}) \geq 1$ | Outliers, Real | $k = 0$ |
| Oscillatory Phase | $\Re(\lambda_{\text{dom}}) \geq 1$ | Outliers, Complex | $k = 0$ |
| Bump Phase | $\Re(\lambda_{\text{dom}}) \geq 1$ | Outliers, Real | $k \neq 0$ |
| Wave Phase | $\Re(\lambda_{\text{dom}}) \geq 1$ | Outliers, Complex | $k \neq 0$ |
| Chaotic Phase | $\Re(\lambda_{\text{dom}}) \geq 1$ | Bulk Disk | - |

### 3.4 The Role of Key Structural Parameters in Shaping the Spectrum

**Excitaion-Inhibition Balance**   The magnitude of excitatory and inhibitory interaction affects the magnitude of the real parts of the outlier eigenvalues, which in turn influences the degree of deviation of neural activity from the fixed point and the level of synchronization. As shown in Eq. 3, the magnitude of excitatory and inhibitory interactions influences the elements of the effective interaction matrix. As shown in Fig. 2(a), the more excitatory interaction there is, the larger the real part of the outlier eigenvalues, the greater the degree to which neural activity deviates from the fixed point, and the more synchronized the neural activity becomes; the opposite is also true.

**Excitation-Inhibition Loop**   The Excitation-Inhibition Loop is considered a crucial component for the emergence of neural oscillations [1, 33]. Our theory explains this from the perspective of the eigenvalue spectrum. As shown in Fig. 2(a), the greater the magnitude of the interaction between excitatory and inhibitory neurons, the more likely it is for the outlier eigenvalues to have an imaginary part, causing neural oscillations to emerge(Eq. 3). This is because $\bar{g}_{IE}$ and $\bar{g}_{EI}$ are located on the off-diagonal elements of the effective interaction matrix, and an increase in $\bar{g}_{IE}$ and $\bar{g}_{EI}$ can lead to the appearance of an imaginary part in the outliers.

**The mismatch of Excitation/Inhibition projection range**   The mismatch of projection range $d_{\alpha\beta}$ of types of neurons causes the emergence of spatial patterns. As shown in Eq. 3, the elements of the effective matrix $\bar{\mathbf{J}}_{\text{eff}}^{(k)}$ contain decay factors $\exp\left(-||k||^2 d_{\alpha\beta}^2/2\right)$. If all the projection ranges $d_{\alpha\beta}$ are the same and let's denote $d_{\alpha\beta} = d$, the eigenvalues would follow the relationship as $\lambda^{(k)} = \lambda^{(0)} \exp\left(-||k||^2 d^2/2\right)$. In this case, only the eigenvalues corresponding to the wave vector $k = 0$ can have the largest real part. However, if the projection range $d_{\alpha\beta}$ of types of neurons is mismatched, eigenvalues corresponding to wave vector $k \neq 0$ can have the largest real part.

A common way for a spatial pattern to emerge is when the projection range of inhibitory neurons is greater than that of excitatory neurons, which is known as lateral inhibition [35, 34]. As shown in the comparison between two figures of Fig. 2(a), this mechanism is also applicable in our model. Our theory explains this from the perspective of the eigenvalue spectrum. This is because the elements of the effective matrix $\bar{\mathbf{J}}_{\text{eff}}^{(k)}$ contain decay factors $\exp\left(-||k||^2 d_{\alpha\beta}^2/2\right)$, and the elements decay faster with $k$ if $d_{\alpha\beta}$ is larger. Therefore, the inhibitory elements of the effective matrix may decay faster than excitatory ones. In some wave vectors $k$, the excitation exceeds the inhibition and the real part of these eigenvalues may exceed 1 and spatial patterns of neural activity emerge.

**Local sparsity**   Local sparsity, rather than the overall sparsity, plays a significant role in chaotic neural activity. In a spatially extended neural network, the number of neurons connected to a given neuron is certainly sparse compared to the total number of neurons. However, Eq. 5 indicates that what truly determines the dynamics is the ratio of the number of connections a neuron has to the number of neurons within its projection range $k_{\alpha\beta}^{out}/(\pi d_{\alpha\beta}^2 \cdot N_\alpha)$, namely the "local sparsity" that plays a role in the radius of the bulk disk part, which is different from "standard sparsity" $k_{\alpha\beta}^{out}/N_\alpha$ in the situation without spatial distribution. This suggests that even under globally sparse conditions, the relative density of local connections between neurons in the brain may be the reason it can generate synchronized activities such as traveling waves. Besides, the concept of "local sparsity" also demonstrates many phenomena in numerical simulation, including both spike-based and rate-based models. In most numerical simulations that produce neural activity with spatial patterns,

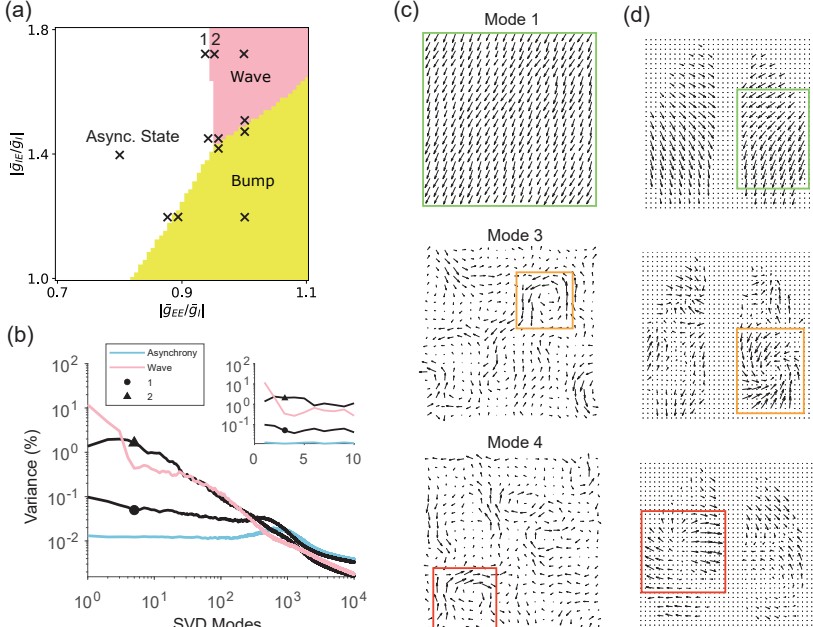

Figure 4: (a) Phase diagram with SVD analysis on combined time series from marked parameter sets. (b) Projections onto SVD modes for: asynchrony center (blue), wave center (pink), and boundaries (circles/triangles). (c) SVD modes 1,3,4 of combined series. (d) Selected experimental SVD modes. Colored boxes highlight matching phase velocity patterns: green (planar waves), orange and red (clock wise & counter clock wise spirals, respectively).

the connection between neurons is often very dense, sometimes even with full connection locally [8, 9, 23], while numerical simulations that produce neural activity without spatial patterns are often sparse [37].

### 3.5 Analyzing Dynamical Transitions

To elucidate phase transitions between distinct dynamical phases, we analyze the spatiotemporal organization of emergent activity patterns. We derive phase velocity fields from network activity to capture the local direction and speed of patterns like propagating waves. By applying Singular Value Decomposition (SVD) to these velocity fields, we identify dominant spatiotemporal modes of activity flow. This allows us to systematically study how these modes reconfigure as the system traverses different dynamical regimes and critical boundaries between them, offering a quantitative window into the nature of these transitions.

Our analysis reveals a key phenomenon at phase boundaries: "mode mixing", where SVD modes characteristic of both adjacent pure phases significantly contribute, indicating dynamically hybrid states consistent with underlying spectral properties near instability (Fig. 4(a-b)). Crucially, the dominant spatial SVD modes identified in our model (e.g., plane waves, spirals) exhibit compelling qualitative similarities to patterns observed in mesoscopic optical imaging of the mouse cortex across different arousal states (Fig. 4(c-d)) [4]. This correspondence suggests our model captures salient principles of spatiotemporal pattern formation and transition relevant to real brain dynamics. See Appendix A.3 for methods and supplementary results.

### 3.6 The corresponding phase of different degrees of consciousness

Having seen the similarity between patterns in our model and the experiment, we aim to establish a relationship between different degrees of consciousness and the corresponding phases in our model. Although the phase might be a multivalued function of brain states (multiple phase patterns may coexist in one brain state), within the phase space we have searched and for a limited number of experimental samples, by comparing order parameters (Fig. 5, see Appendix A.9 for details), we

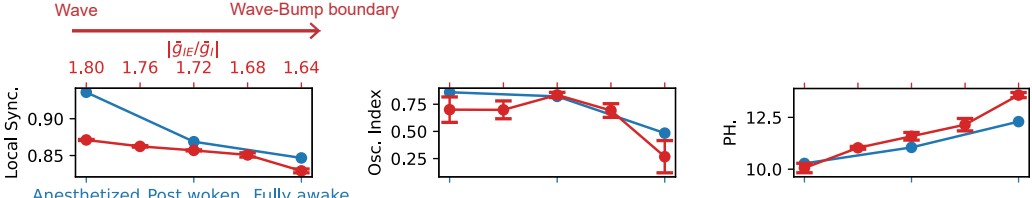

Figure 5: Blue curves represent order parameters of different degrees of consciousness of the same mouse (1 trial). Red curves represent order parameters of parameter sets in the wave phase (5 trials) with $|\bar{g}_{EE}/\bar{g}_I| = 1.04$ and $|\bar{g}_{IE}/\bar{g}_I|$ varying from from the wave phase to wave-bump phase boundary.

found preliminary evidence that from the anesthetized to the fully awake state (increasing degree of consciousness), the corresponding E-to-I coupling strength decreased, moving from the interior of the wave phase towards the wave-bump phase boundary.

This aligns well with [8], which discovered that stronger E-to-I coupling in a spatially extended network led to propagating waves (corresponding to the wave phase in our model), while a weaker one led to bump phase. Pure propagating waves in [8] could not be modulated by external stimuli and had a lower decoding accuracy, aligning with the physiological properties of the anesthetized state. At the wave-bump phase boundary, the network entered a critical state where the modulation effect of stimuli was maximized, aligning with the fully awake state.

## 4    Conclusions and Discussions

We introduced a spectral theory for spatially extended neural networks, quantitatively linking local connectivity (E/I projection ranges, strengths, and local sparsity) to global dynamics via the eigenvalue spectrum of the connectivity matrix. This spectral blueprint, characterized by outlier modes and a bulk disk, accurately predicts a rich repertoire of emergent behaviors including asynchronous states, oscillations, bumps, waves, and chaos, providing a mechanistic bridge from structure to dynamics.

Besides, unlike classical neural field models[12–14], our approach makes no assumption of homogeneous connectivity or the continuum limit, enabling the emergence of chaotic dynamics that traditional neural field theory cannot capture. While such chaotic regimes have been extensively characterized in non-spatial networks using dynamical mean-field theory[19–21], their counterparts in spatially structured systems remain largely unexplored. Our RMT-based analysis bridges this gap, providing a unified and elegant perspective: the outlier eigenvalues correspond to Fourier modes, as in neural field theory, whereas the bulk spectrum reflects DMFT-like statistics. This connection highlights how RMT can serve as a powerful theoretical lens for integrating spatial structure and randomness in large-scale neural dynamics.

This framework contextualizes how observed brain dynamics, such as state-dependent patterns [4], arise. It aligns with neural field theories [14] regarding pattern formation and offers refined insights into chaos generation compared to globally coupled models [19], highlighting the role of local density. Crucially, we posit that our model's static connectivity represents time-varying effective connectivity in the brain, which is constantly reshaped by neuromodulation, stimuli, and attention [38, 39]. Thus, the identified dynamical phases might offer a new perspective: they could serve as candidate states within a larger phase space that the brain potentially traverses, possibly corresponding to different states of consciousness—an idea open to experimental testing.

**Key limitations guide future work**   Extending the theory to nonlinear neural/node's dynamics [21] and incorporating synaptic plasticity to model adaptive spectral changes and learning [40] are paramount. Addressing structural complexities beyond isotropic connectivity, modeling the explicit time-variance of effective connectivity, and robustly inferring spectral features from empirical data [41] are also critical. Also, a more direct characterization of chaos needs to be done in future works. Furthermore, elucidating the direct computational roles of these spectrally-defined dynamical regimes remains a vital pursuit [42]. Besides, multiple phase patterns may superimpose or coexist simultaneously due to nonlinear mode-coupling, which need to be further investigated.

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

# A  Appendixes

## A.1  Supplementary discussion

Our calculation of the bulk disk radius builds upon foundational random matrix theory concerning the circular law, applicable even to non-Gaussian weight distributions [26], and methods for structured random networks [30, 29, 28]. The emergence of outlier eigenvalues from low-rank perturbations to such random matrices is also well-established [27]. Our novelty lies in systematically deriving both these spectral components for spatially extended E/I networks and explicitly linking them to distinct dynamical phases. While prior work has explored outlier eigenvalues in non-spatial networks, often focusing on single global outliers [43], population modes [44] without considering spatial, or randomly distributed local outliers that can be removed by zero-sum constrain[30, 45], and while some studies noted the presence of spatially-organized eigenvalues as a secondary observation without direct calculation or dynamic linkage [46], our framework uniquely connects the full set of spatially-indexed outlier eigenvalues (for wave vectors) and the bulk disk radius to the emergence of diverse spatiotemporal patterns and chaos, respectively.

The explicit calculation of eigenvalues associated with specific wave vectors is crucial for understanding spatially patterned activity. While some models of cortical networks have implicitly or explicitly involved such wave numbers, they often relied on simplifications from neural field theory for analytical tractability [18, 17, 23] or focused on conditions for specific instabilities, such as how E/I imbalance can lead to pattern formation at particular wave numbers [15]. Our approach provides a more general matrix-based spectral method that directly ties the parameters of the spatially explicit network (projection ranges, strengths) to the entire outlier spectrum without necessarily reducing to a continuum field limit, thereby offering a direct bridge from discrete network structure to emergent spatial dynamics like bumps and waves.

## A.2  Experimental Data of different degrees of consciousness of mice

The experiment [4] (https://doi.org/10.5281/zenodo.7574791) used mesoscopic optical imaging of mice expressing a genetically encoded voltage indicator in cortical pyramidal neurons, to access spontaneous population voltage activity across both hemispheres of the dorsal cortex. "Anesthetized" refers to that the mice underwent light anesthesia induced by a bolus injection of pentobarbiturate. "Post woken" refers to that the mice woke up from anesthesia as indicated by occasional spontaneous coordinated whisker and body movements. "Fully awake" refers to that the mice were well habituated to the imaging conditions and had been free of anesthesia for at least 3 days prior to the imaging session.

## A.3  Phase Velocity Field analysis and Wave Pattern Detection

**Signal Pre-processing for Phase Extraction:**   To analyze wave patterns from time-series data recorded at multiple spatial sites (e.g., from optical imaging or electrophysiological arrays), we first extract the instantaneous phase from each recording site. This process involves two main steps:

1. **Band-pass Filtering:** The raw signal from each site, denoted as $s_{raw}(t)$, is first band-pass filtered to isolate activity within a frequency range of interest. This step serves to remove high-frequency noise and focus the analysis on specific neural oscillations (e.g., delta: 0.5-4 Hz; theta: 4-8 Hz [47]; alpha: 8-13 Hz) or on frequency bands containing significant signal power as determined by power spectral analysis of the recordings. The choice of frequency band is also constrained by the temporal resolution of the recording technique. Let the filtered signal be $x(t)$.

2. **Phase Extraction via Hilbert Transform:** The instantaneous phase $\phi(t)$ is extracted from the filtered signal $x(t)$ at each site using the Hilbert transform. The analytic signal $z(t)$ is constructed as,
$$z(t) = x(t) + i\mathcal{H}\{x(t)\} = A(t)e^{i\phi(t)}, \tag{7}$$
where $\mathcal{H}\{x(t)\}$ is the Hilbert transform of $x(t)$, $A(t)$ is the instantaneous amplitude, and $i$ is the imaginary unit. The instantaneous phase $\phi(t)$ is then obtained as,
$$\phi(t) = \arctan2(\mathcal{H}\{x(t)\}, x(t)), \tag{8}$$

where $\mathrm{arctan2}(y, x)$ is the two-argument arctangent function that correctly resolves the phase into all four quadrants. The result of this pre-processing is a phase time series $\phi_j(t)$ for each spatial recording site $j$.

**Optical Flow Method for Phase Velocity Field Estimation:** To quantify the propagation of phase patterns across the spatial recording array, we employ an optical flow method, specifically the Horn-Schunck algorithm, adapted for phase data. This method estimates a 2D velocity field $(v_x(x, y, t), v_y(x, y, t))$ that describes the motion of surfaces of constant phase.

Let $I(x, y, t) = \phi(x, y, t)$ represent the instantaneous phase at spatial location $(x, y)$ and time $t$. The core assumption of optical flow is brightness constancy, which for phase translates to phase constancy along a trajectory: $I(x + v_x dt, y + v_y dt, t + dt) = I(x, y, t)$. A first-order Taylor expansion yields the optical flow constraint equation:

$$I_x v_x + I_y v_y + I_t = 0, \tag{9}$$

where $I_x = \frac{\partial I}{\partial x}$, $I_y = \frac{\partial I}{\partial y}$, and $I_t = \frac{\partial I}{\partial t}$ are the spatial and temporal partial derivatives of the phase field.

To solve for the two unknown velocity components $(v_x, v_y)$ from this single equation, the Horn-Schunck method introduces a global smoothness constraint, minimizing an energy functional $E$:

$$E = \iint \left[ (I_x v_x + I_y v_y + I_t)^2 + \alpha^2 (\|\nabla v_x\|^2 + \|\nabla v_y\|^2) \right] dx dy, \tag{10}$$

where $\alpha^2$ is a regularization parameter that weights the smoothness term. Minimization of this functional leads to a system of coupled partial differential equations for $v_x$ and $v_y$. Discretizing these equations (e.g., using finite differences for derivatives and the Laplacian $\nabla^2$) results in a large system of linear equations:

$$I_x (I_x v_x + I_y v_y + I_t) - \alpha^2 \nabla^2 v_x = 0, \tag{11a}$$

$$I_y (I_x v_x + I_y v_y + I_t) - \alpha^2 \nabla^2 v_y = 0, \tag{11b}$$

where $v_x$ and $v_y$ now represent values at discrete grid points $(x_i, y_j)$. The Laplacian terms $\nabla^2 v_x$ and $\nabla^2 v_y$ are typically approximated using a five-point stencil, e.g., $\nabla^2 v_x(x, y) \approx \frac{\overline{v_x}(x,y) - v_x(x,y)}{(\Delta x/2)^2}$, where $\overline{v_x}(x, y)$ is the average of $v_x$ at the four cardinal neighbors of $(x, y)$, and $\Delta x = \Delta y$ is grid spacing.

This system is solved iteratively for each time step $t$:

$$v_x^{(k+1)} = \overline{v_x}^{(k)} - \frac{I_x (I_x \overline{v_x}^{(k)} + I_y \overline{v_y}^{(k)} + I_t)}{I_x^2 + I_y^2 + \lambda_s^2}, \tag{12a}$$

$$v_y^{(k+1)} = \overline{v_y}^{(k)} - \frac{I_y (I_x \overline{v_x}^{(k)} + I_y \overline{v_y}^{(k)} + I_t)}{I_x^2 + I_y^2 + \lambda_s^2}, \tag{12b}$$

where $(k)$ denotes the iteration number, and $\lambda_s^2$ (related to $\alpha^2$ and grid spacing $\Delta x, \Delta y$, e.g., $\lambda_s^2 \approx (2\alpha/\Delta x)^2$) encapsulates the smoothness constraint. Iterations proceed until convergence or for a fixed number of steps.

**Implementation Details:**

- **Partial Derivatives:** Spatial derivatives $I_x, I_y$ were computed from the phase maps $\phi(x, y, t)$ at each time $t$ using a Sobel filter or a five-point central difference scheme, averaged between two consecutive time frames $t$ and $t + dt$. The temporal derivative $I_t$ was computed using a forward difference between $\phi(x, y, t + dt)$ and $\phi(x, y, t)$, potentially after local spatial averaging to reduce noise.

- **Boundary Conditions:** For calculating spatial derivatives near boundaries, appropriate schemes (e.g. Neumann boundary conditions where derivatives are zero) were applied. For the averaging terms $\overline{v_x}, \overline{v_y}$ in the iterative update, Neumann or zero-padding (Dirichlet-like) boundary conditions were typically used for the velocity components outside the defined spatial grid. The specific choice of derivative computation and boundary conditions was validated on test data to ensure reasonable velocity fields.

- **Regularization Parameter $\alpha$:** The value of $\alpha$ (or $\lambda_s$) was chosen empirically to balance adherence to the optical flow constraint with the smoothness of the resulting velocity field, often by visual inspection of results on synthetic or sample data.

This procedure yields a phase velocity field $(v_x(x, y, t), v_y(x, y, t))$ for each time point, which can then be further analyzed to characterize wave properties like direction, speed, and coherence.

**Supplementary Results of Phase Velocity Field Analysis** Our spectral understanding of how network structure dictates distinct dynamical phases can also illuminate the nature of transitions between these phases. To quantitatively characterize these transitions, particularly at phase boundaries and triple points where multiple dynamical tendencies may coexist or compete, we analyze the spatiotemporal structure of phase velocity fields. Analyzing the instantaneous phase $\phi(x, y, t)$ offers several advantages for understanding organized spatiotemporal patterns. The phase captures the relative timing of oscillatory activity across different spatial locations, making it particularly sensitive to propagating waves, synchronized domains, and their complex interactions. By then computing the optical flow of these phase maps, we obtain a velocity field $(v_x(x, y, t), v_y(x, y, t))$ that directly quantifies the local direction and speed of these emergent patterns. This provides a rich, time-varying representation of the network's collective spatiotemporal organization, which is amenable to techniques like Singular Value Decomposition (SVD) for identifying dominant modes of activity flow.

The methodology for obtaining phase velocity fields from neural activity (either model-generated or experimental) involves band-pass filtering (e.g., 0-40 Hz, based on signal power spectrum), Hilbert transform to extract instantaneous phase $\phi(x, y, t)$, and an optical flow algorithm to compute the velocity vectors of phase propagation. Full details of this pre-processing and optical flow computation are provided in Appendix A.3.

To analyze the structure of these time-varying phase velocity fields $\mathbf{V}_{\text{field}}(x, y, t)$, we employ SVD. The velocity field data across all spatial sites (both $v_x$ and $v_y$ components) and time points is arranged into a matrix $\mathbf{X}$, where rows typically represent time and columns represent flattened spatial components. SVD decomposes this matrix as $\mathbf{X} = \mathbf{U}\boldsymbol{\Sigma}\mathbf{V}^{\dagger}$, where columns of $\mathbf{U}$ are temporal modes, columns of $\mathbf{V}$ are spatial modes, and $\boldsymbol{\Sigma}$ contains the singular values indicating the contribution of each mode.

To study phase transitions, we first identify a set of common spatial modes by performing SVD on a combined dataset of phase velocity fields from parameter sets spanning different phases, phase boundaries, and triple points (see markers in Fig. 4a). Then, for each individual parameter set (numbered markers at a phase boundary in Fig. 4a), its phase velocity field time series is projected onto these common spatial modes. The resulting projection weights quantify how much each common spatial mode contributes to the dynamics of that specific parameter set. By examining the profile of these projection weights (variance explained by each mode) for parameter sets systematically chosen along a path crossing a phase boundary, we can characterize the transition.

Fig. 4b illustrates this for points near the asynchrony-wave phase boundary. Parameter sets located on a phase boundary exhibit projection profiles that are hybrid, sharing features with the profiles of the pure phases they separate. SVD modes characteristic of both pure asynchrony and pure wave states contribute significantly. At a triple point, the dynamics reflect a richer mixture, with contributions from modes associated with all three converging phases (see Fig.6). This suggests that at these critical junctures in parameter space, the system's dynamics are not committed to a single attractor but can explore or blend features of multiple underlying dynamical regimes.

This "mode mixing" at boundaries and triple points can be intuitively understood from our spectral theory. Near these critical regions, the eigenvalue spectrum of the connectivity matrix may exhibit near-degeneracies, where multiple eigenvalues (corresponding to different potential dynamical patterns, e.g., a $k = 0$ oscillatory mode and a $k \neq 0$ wave mode) have comparable real parts close to the instability threshold. Small amounts of perturbation can then cause the system to fluctuate between, or simultaneously express aspects of, these competing dynamical modes. This is consistent with experimental observations where brain activity can show transient or mixed features, especially during state transitions [48, 49].

Encouragingly, the dominant SVD spatial modes extracted from our model's phase velocity fields, such as plane waves or spiral patterns (Fig. 4c), show qualitative similarities to modes extracted

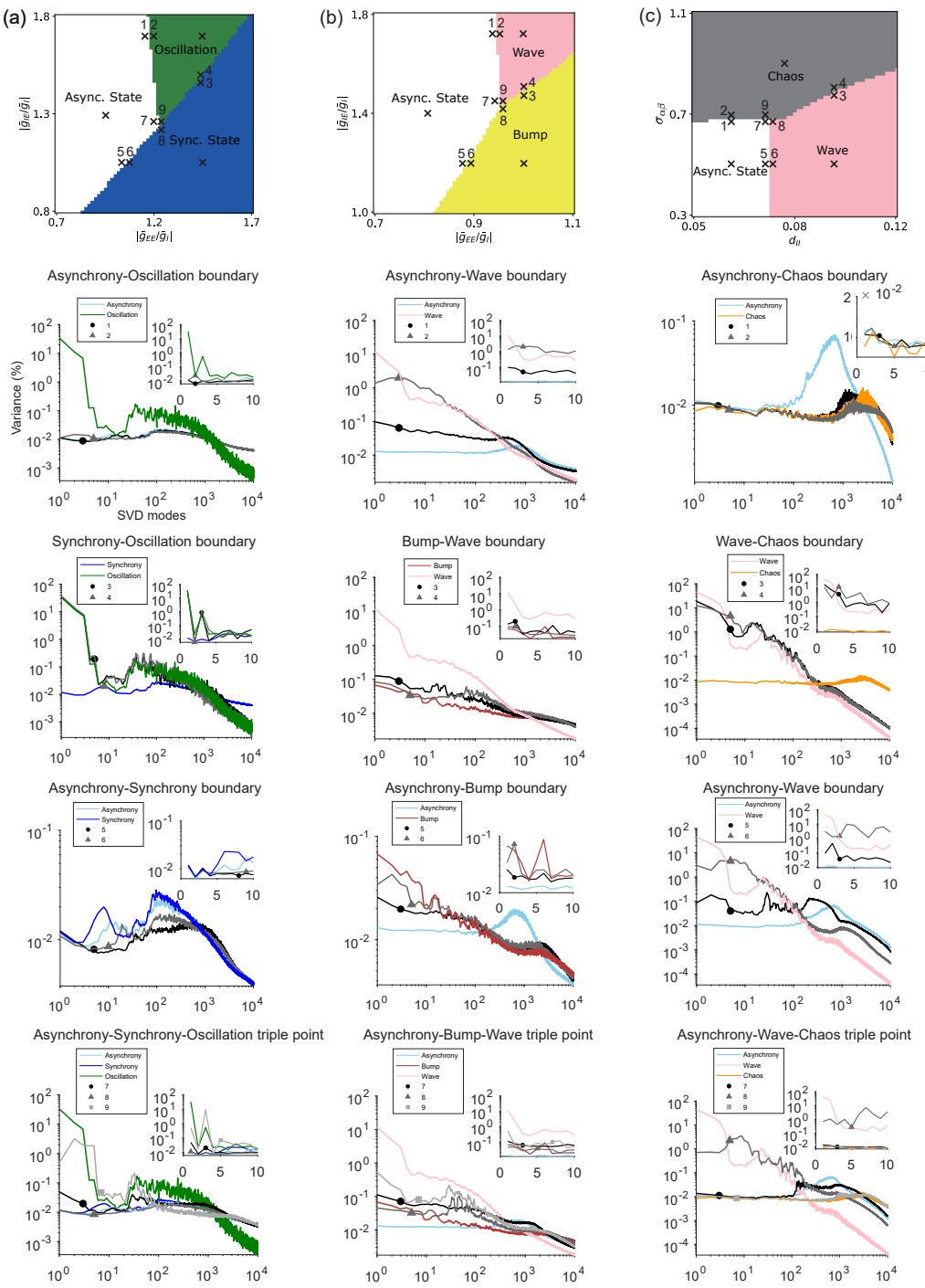

Figure 6: Analysis of all the phase boundaries and triple points. (a) Asynchrony-bump-wave phase diagram, and projection curves onto SVD modes of the combined time series of the parameter sets marked by crossings in the phase diagram. Sky blue line refers to asynchrony center, green line refers to oscillation center, dark blue line refers to synchrony center. Circles, triangle and squares refers to the corresponding parameter sets marked by numbers in the phase diagram. The inset shows the projection curves onto the first ten modes. (b) Same as (a), pink line refers to wave center, dark red line refers to bump center. (c) Same as (a), orange line refers to chaos center.

from mesoscopic optical imaging of mouse cortex [4] across different arousal states (Fig. 4d). This suggests that the underlying principles governing the formation and transition of spatiotemporal patterns in our model may capture salient aspects of real brain dynamics.

## A.4   Order Parameters

In order to numerically validate the correctness of the prediction of phase. We introduce order parameters **CV.** (Coefficient of Variation) and **Mean Acti.** (Mean Activity) to detect the magnitude of the neural activity fluctuation, introduce **Osc. index**(Oscillation Index) to detect the neural oscillation, introduce **Local Sync.** (Local Synchronization) and **Moran's Index** to detect the spatial patterns of neural activity.

**Mean Acti.** (Mean Activity) This order parameter calculates the magnitude of neurons' firing rate. This order parameter is useful for neural networks with hyperbolic tangent activation functions, given by

$$Mean\ Acti. = \left\langle |\phi(h_i^E)| \right\rangle_i .$$

**CV.** (Coefficient of Variation) This order parameter describes the magnitude of neural activity fluctuation. It calculates the ratio between the standard deviation and mean of the neural firing rate. This order parameter is useful for neural networks with rectified linear, supra-linear and etc. activation functions:

$$CV. = \left\langle \frac{Var(\phi(h_i^E))}{\mathbb{E}(\phi(h_i^E))} \right\rangle_i .$$

**Osc. Index** (Oscillation Index) This parameter describes the magnitude of neural oscillation. Similar to [50], we define this order parameter as the fraction of the Fourier spectrum energy of the peak concentrated at oscillation frequency. The Fourier spectrum is calculated by averaging the Fourier spectrum of all the excitatory neurons.

**Local Sync.** (Local Synchronization) This parameter describes the degree of local synchronization of neural activity. It calculates the ratio between the mean of the firing rate and the mean of the absolute value of the firing rate of local neurons. We define local neurons as the neurons within a square with a side length of 10 times the inter-neuron spacing. We chose this side length so that it's similar to the detection range of local field potential. This order parameter is useful for neural networks with hyperbolic tangent activation functions:

$$Local\ Sync. = \left\langle \frac{\langle \phi(h_i^E(t)) \rangle_{i,\ local}}{\langle |\phi(h_i^E(t))| \rangle_{i,\ local}} \right\rangle .$$

**Moran's Index** This parameter detects the existence of spatial patterns. It calculates the ratio between the mean of correlation between local neurons and the mean of correlation between all neurons. The definition of "local neurons" is the same to Local Sync:

$$Moran'\ Index = \left\langle \frac{\sum_{i=1}^{N_E} (\phi(h_i^E(t)) - \overline{\phi(h^E(t))}) \sum_{j \in nb.} (\phi(h_j^E(t)) - \overline{\phi(h^E(t))})}{N_{nb.} \sum_{i=1}^{N_E} (\phi(h_i^E(t)) - \overline{\phi(h^E(t))})^2} \right\rangle_t .$$

**PH.** (Persistent Homology) This parameter is used to characterize the degree of spatial localization in a 2D scalar field $f(x, y)$. The method is to set a threshold $t$ and increase it from the minimum to the maximum value of $f(x, y)$, and then recognize connected components in the sublevel sets $X_t = \{(x, y) | f(x, y) \leq t\}$ at each threshold $t$. As we sweep through $t$, new connected components (blobs) are born, and existing ones merge or vanish. Each such event is recorded as a birth-death pair $(b_i, d_i)$, representing the threshold at which a component appears and disappears respectively. The lifetime of a blob is defined as $d_i - b_i$. The persistent homology of $f(x, y)$ is defined as the sum of the lifetime of all the blobs. Larger persistent homology means a higher degree of pattern localization. Persistent homology is independent of the size of the field (number of grid points here), and is only dependent on the spatial pattern and the contrast of the pattern.

Before calculating PH., we first smooth the image for both model and experiment using *ndimage.generic_filter* from *scipy*. The kernel size is $20 \times 20$ and $2 \times 2$ for model and experiment respectively, proportional to their field size ($200 \times 200$ for model, $\sim 20 \times 40$ for each hemisphere

of the cortex). After smoothing, we take the z-scored image by subtracting the mean from it and then dividing by the standard deviation. When calculating PH. of the z-scored image, we remove the birth-death pairs whose lifetime is less than 2% of the range of the z-scored image to further remove high-frequency noise. Specifically, for the experimental data, we first take the z-scored image with two hemispheres as a whole, and then calculate PH. separately for each hemisphere and finally take the mean value.

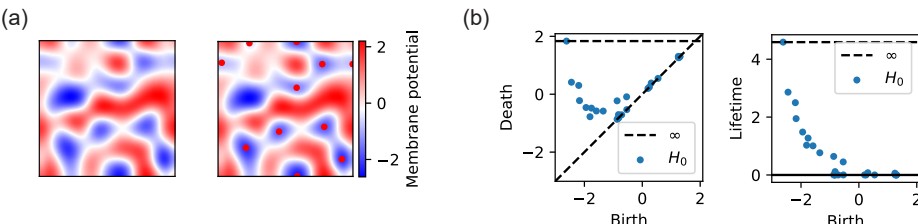

Figure 7: (a) Smoothed network pattern of the center of the wave phase, marked by black star in Fig. 2. (b) Red dots represent blobs found. (c) Death and birth values of blobs. (d) Lifetime of blobs.

## A.5 Numerical Simulation

For the theoretical prediction of phase, we predict $61 \times 61$ gird points for a single phase diagram. For numerical simulation of phase prediction, the grid points is $21 \times 21$. For each grid point, we independently initialize the connectivity matrix and conduct 5 times simulations with a total simulation time of 100 times of time constant of membrane potential $\tau$ and a step length of $0.01\tau$. We only begin to calculate the order parameters after $25\tau$ of simulation in order to avoid the influence of the transient process. Our numerical experiments were conducted on a computing cluster consisting of 16 nodes, each equipped with an Intel(R) Xeon(R) CPU E5-2407 0 @ 2.20GHz.

## A.6 Numerical Experiments with Alternative Activation Functions

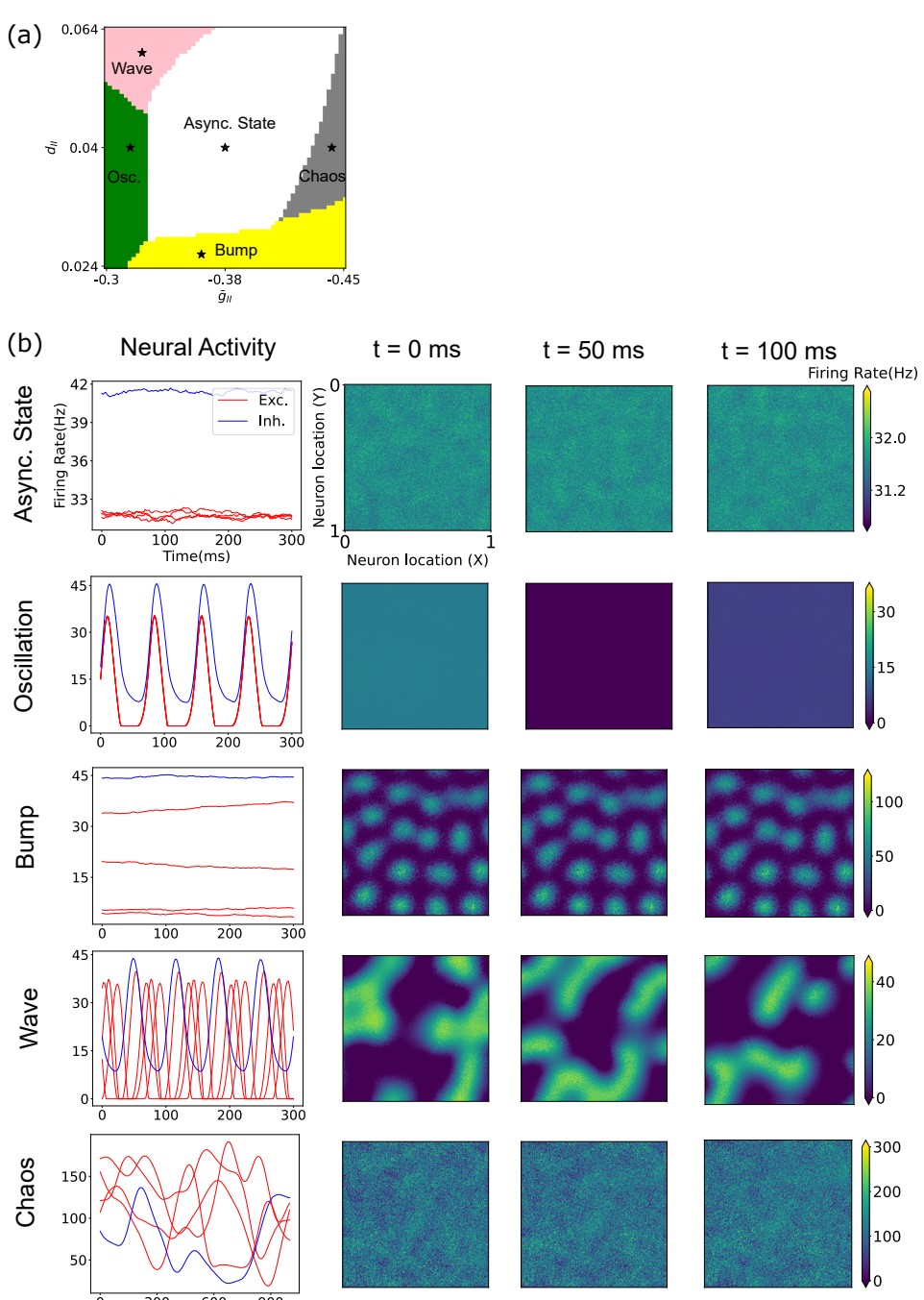

Figure 8: Dynamical regimes in spatially extended networks with alternative activation functions. (a) The Phase diagram under alternative activation functions. (b) Representative neural activity and spatial patterns of excitatory neurons in different dynamical phases over time.

## A.7 Numerical Results of Order Parameters and Phase Diagrams

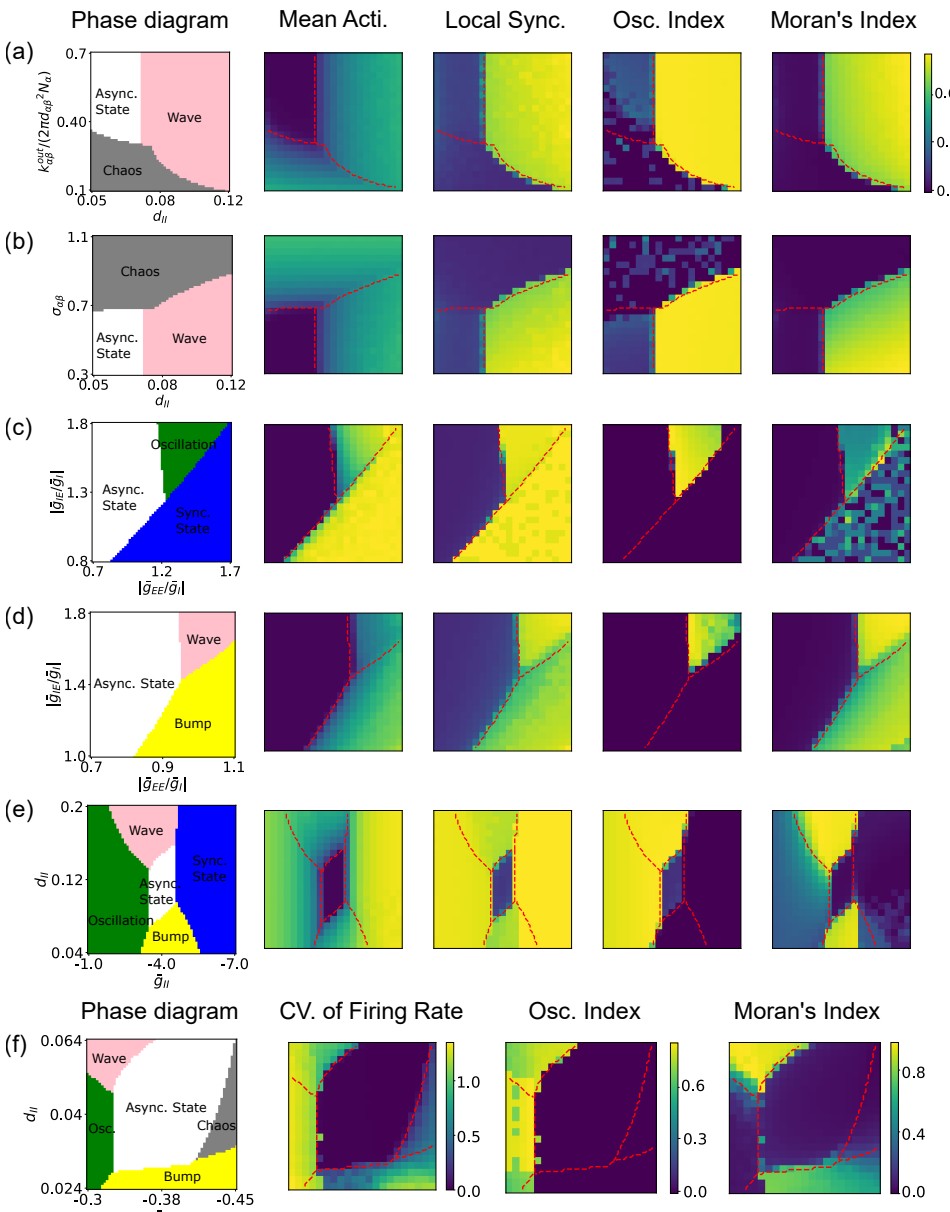

Figure 9: Phase diagrams and order parameters under different parameters.

## A.8 Numerical Results of Order Parameters Calculated Using Membrane Potential

When comparing order parameters of the model and the experiment, because the experimental data are the spontaneous population membrane voltage fluctuations of pyramidal neurons, we should correspondingly use the membrane potential of excitatory neurons when calculating order parameters of our model. The results about order parameters above are calculated using neuron activity (by applying an activation function to the membrane potential), and below in Fig. 10 are results calculated using membrane potential of excitatory neurons.

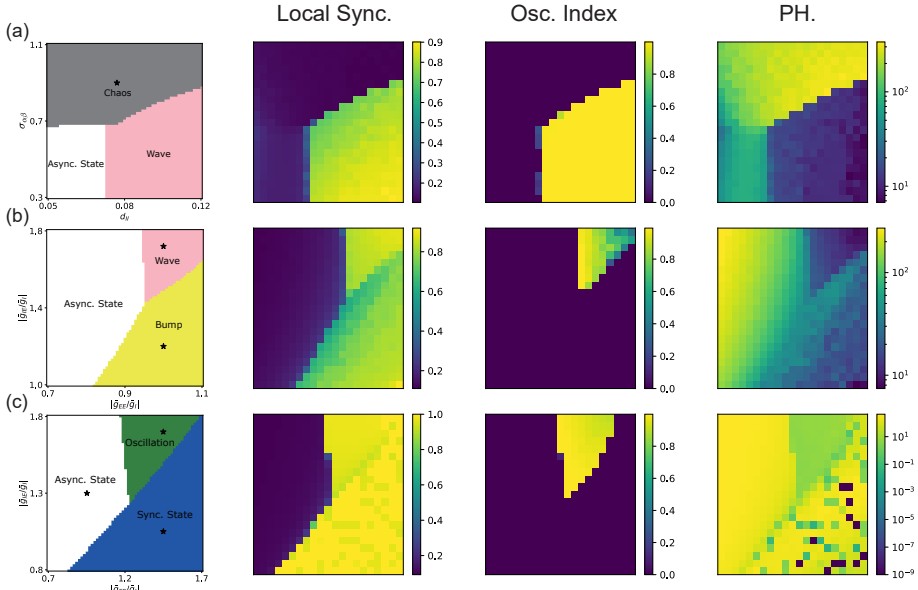

Figure 10: Order parameters calculated with membrane potential.

## A.9   Find the Corresponding Phase of Different Degrees of Consciousness

Firstly, we can determine which phases the experimental states are in by comparing order parameters such as local sync. and osc. index of our model and the experiment. As is shown in Fig. 5 and Fig. 10, all three states of the experiment have high local sync.; thus we can exclude the asynchrony and chaos phases. By comparing osc. index, we can exclude the bump and synchrony phases. To distinguish between the oscillation and wave phases, we do not use Moran's index as before because the essential difference between unconscious and conscious states as reflected in Fig. 1 is the degree of localization of patterns, which cannot be characterized by Moran's index. Instead, we use PH. (see Appendix A.4), which can quantify the degree of pattern localization and distinguish between the oscillation and wave phases at the same time. The oscillation phase has PH. generally lower than all states of experiment, thus can also be excluded. To conclude, the three consciousness states are all in the wave phase.

Next we try to identify their difference of location in the wave phase. In the region of the wave phase we searched in the phase diagram varying $d_{II}$ and $\sigma_{\alpha\beta}$ (Fig.10(a)), the osc. index is too high to be compatible with the experiment. Therefore we confine our region of interest to the wave phase in the phase diagram varying $|\bar{g}_{IE}/\bar{g}_I|$ and $|\bar{g}_{EE}/\bar{g}_I|$ (Fig.10(b)). We cannot find the exact coordinate of a brain state in the phase diagram, because the correspondence between network structural parameters and the order parameters we used is not a bijection. Instead, each brain state corresponds to a region in the phase diagram. In addition, the precise region location cannot be determined because our grids are not dense enough. Therefore we are only concerned with the changing trend in the phase diagram from the anesthetized to the fully awake state.

We can find a trajectory from the interior of the wave phase to the wave-bump phase boundary, as is shown in Fig. 5, where the changing trends of local sync., osc. index and PH. are all similar between model and experiment. The corresponding varying structure parameter is $|\bar{g}_{IE}/\bar{g}_I|$, decreasing from unconscious to conscious state, indicating that a higher degree of consciousness is associated with a smaller coupling strength from excitatory to inhibitory neurons.

As for the limitation that a precise corresponding region location cannot be determined, future research can utilize more order parameters to narrow down the possible region of brain states on the phase diagram, and use denser grids to determine the exact region location. Additionally, our experimental results are limited to one trial of the same mouse (there is only one mouse who has data of all three degrees of consciousness in the open source dataset introduced in A.2). Future validation on different datasets with more mice and trials remains to be done.

## A.10 Parameters of Phase Diagrams

| Parameter | Fig. 9(a) | Fig. 9(b) | Fig. 9(c)/2(a) | Fig. 9(d)/2(a) | Fig. 9(e) | Fig. 9(f) |
|---|---|---|---|---|---|---|
| $N_E$ | 40000 | 40000 | 40000 | 40000 | 10000 | 40000 |
| $N_I$ | 10000 | 10000 | 10000 | 10000 | 2500 | 10000 |
| $k_{EE}^{out}$ | 62.8-439.8 | 596.90 | 1169.93 | 596.90 | 596.90 | 321.70 |
| $k_{IE}^{out}$ | 15.7-110.0 | 149.23 | 292.48 | 149.23 | 149.23 | 80.42 |
| $k_{EI}^{out}$ | 62.8-439.8 | 596.90 | 1169.93 | 1169.93 | 596.90 | 321.70 |
| $k_{II}^{out}$ | 15.7-110.0 | 149.2-859.5 | 292.48 | 292.48 | 23.9-596.9 | 29.0-205.9 |
| $d_{EE}$ | 0.05 | 0.05 | 0.07 | 0.05 | 0.10 | 0.04 |
| $d_{IE}$ | 0.05 | 0.05 | 0.07 | 0.05 | 0.10 | 0.04 |
| $d_{EI}$ | 0.05 | 0.05 | 0.07 | 0.07 | 0.10 | 0.04 |
| $d_{II}$ | 0.05-0.12 | 0.05-0.12 | 0.07 | 0.07 | 0.0-0.2 | 0.0-0.1 |
| $\bar{g}_{EE}$ | 5.50 | 5.50 | 6.3-15.3 | 6.3-9.9 | 5.50 | 0.57 |
| $\bar{g}_{IE}$ | 5 | 5 | 7.2-16.2 | 9.0-16.2 | 5 | 0.12 |
| $\bar{g}_{EI}$ | -5 | -5 | -9 | -9 | -5 | -1.90 |
| $\bar{g}_{II}$ | -4.25 | -4.25 | -9 | -9 | -1.0 - -7.0 | -0.3 - -0.5 |
| $\sigma_{EE}$ | 0.55 | 0.3-1.1 | 0.10 | 0.10 | 0.10 | 0 |
| $\sigma_{EI}$ | 0.55 | 0.3-1.1 | 0.10 | 0.10 | 0.10 | 0 |
| $\sigma_{IE}$ | 0.55 | 0.3-1.1 | 0.10 | 0.10 | 0.10 | 0 |
| $\sigma_{II}$ | 0.55 | 0.3-1.1 | 0.10 | 0.10 | 0.10 | 0 |

## A.11 Relation between Eigenvalues and Dynamics

We want to understand the relationship between connectivity structure and dynamics of spatial distributed neural networks. To begin with, we first consider a simple linear neural network, of which dynamics follows:

$$\frac{dh_i}{dt} = -h_i + \sum_j J_{ij} h_j + \xi_i(t). \tag{13}$$

Because it's a linear dynamical system, the dynamics can be decomposed into different independent modes. Let's assume the connectivity matrix $J$ is diagonalizable. $J = A\Lambda A^{-1}$, where $A$ is composed of eigenvectors of connectivity matrix, $A = [\nu_1, \nu_2, \cdots, \nu_N]$, and $\Lambda = [\lambda_1, \lambda_2, \cdots, \lambda_N]$ is composed of eigenvalues of connectivity matrix. The dynamical equation can be rewritten as,

$$\frac{d}{dt}(A^{-1}h) = -A^{-1}h + \Lambda A^{-1}h + A^{-1}\xi_i(t). \tag{14}$$

Therefore, we can consider the dynamics of each eigenvector component to be independent. The activity of neurons is a superposition of components in different directions. Let $h(t) = \sum_i c_i(t)\nu_i$. $c_i$ is the magnitude of independent components, which satisfies,

$$\frac{dc_i}{dt} = (\lambda_i - 1)c_i + A^{-1}\xi(t). \tag{15}$$

For a component with $Re(\lambda_i) < 1$, the magnitude is bounded and fluctuates around $0$. For a component with $Re(\lambda_i) \geq 1$, the magnitude increases over time as a rate of $e^{Re(\lambda_i)-1}$. Therefore, the component with an eigenvalue with the largest real part dominates the dynamics of neural networks.

For a non-linear neural network, we can also use eigenvalues to understand the relationship between connectivity structure and eigenvalues. We only need to perform a linear expansion around the fixed point of neural activity. Let's denote the fixed point of membrane potential as $h^*$, and the deviation from the fixed point as $\delta h$. The neural activity follows,

$$\frac{dh_i}{dt} = -h_i + \sum_j J_{ij}\phi(h_j) + \xi_i(t). \tag{16}$$

We can perform a linear expansion around the fixed point $h^*$. The deviation $\delta h$ from fixed point follows,

$$\frac{d}{dt}\delta h_i = -\delta h_i + \sum_j J_{ij}\phi'(h_j^*)\delta h_j + \xi_i(t). \tag{17}$$

Therefore, we can consider a nonlinear neural network equivalent to a linear neural network with an effective connectivity matrix $\widetilde{J}_{ij} = J_{ij}\phi'(h_j^*)$.

The dynamical regime of neural networks can be characterized through spectral analysis of the effective connectivity matrix $\widetilde{J}_{ij}$. When all eigenvalues satisfy $Re(\lambda_i) < 1$, neural activity remains bounded near the fixed point, exhibiting small-amplitude fluctuations around the steady-state firing rate. This regime corresponds to asynchronous irregular activity due to the absence of dominant eigenmodes.

When spectral outliers emerge with $Re(\lambda_i) \geq 1$, the corresponding eigenmodes dominate network dynamics. These regimes can be systematically classified (Table 1) based on two spectral properties of the dominant eigenvalues: 1) temporal frequency (real vs. complex eigenvalues) and 2) spatial frequency (wave vector $k$ of eigenvectors). Spatial organization in eigenvectors generates distinct spatiotemporal patterns, enabling classification into four phases: synchronized state, oscillatory phase, bump attractor, and traveling wave.

In contrast, when dominant eigenvalues reside within the bulk spectral disk, the system enters a chaotic phase characterized by: (i) spatially unstructured eigenvectors, (ii) large-amplitude fluctuations, and (iii) weak inter-neuronal correlations. Intuitively, the absence of spatial patterning in neural activity stems from the structural homogeneity of dominant spectral bulk eigenmodes. Weak inter-neuronal correlations emerge from high-dimensional superposition of components satisfying $Re(\lambda_i \geq 1)$. While spectral analysis provides initial insights, analytical determination of phase boundaries requires dynamical mean-field theory, as detailed in Appendix A.14.

This method is not fully mathematically rigorous. First, it cannot fully deal with the case of multiple fixed points. Second, it cannot fully predict the dynamic behavior when the network activity goes far from a fixed point. Besides, it cannot fully characterize the dynamic behavior where the fixed point is heterogeneous between different neurons.

Therefore, to verify the validity of our theory under nonlinear conditions, we performed many numerical experiments as aforementioned to demonstrate that our theory is indeed correct under nonlinear conditions and is useful and informative in understanding the relationship between neural networks' dynamics and their connectivity structure.

## A.12   Neural Activity of the Asynchronous State

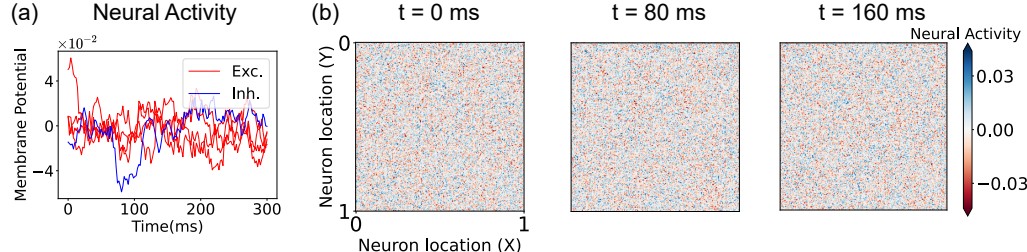

Figure 11: Neural activity of asynchronous state. (a) temporal evolution of membrane potentials of excitatory neurons. (b) spatial patterns of excitatory neurons' firing rate.

As illustrated in Fig. 11, neural populations in the asynchronous state exhibit weak pairwise correlations, resulting in the absence of emergent spatial patterns. Recent theoretical advances by [25], however, demonstrate that low-rank connectivity structures can induce spectral outliers in the long-time window covariance matrix. This finding motivates systematic investigation of spectral properties of covariance matrices in spatially extended neural networks – a promising direction for future research.

Notably, weak deviations from the fixed point emerge through external input modulation. In our experimental paradigm, network input originates from independent white noise sources with low

amplitude, resulting in small deviations from the fixed point. The neural activity under other types of external input needs to be further explored.

### A.13 Eigenvalues and Eigenvectors of Spatially Distributed Neural Networks

### A.13.1 Circular Law

A classical result of the random matrix is the circular law. If you take an $n \times n$ matrix with independent and identically distributed (i.i.d.) entries $J_{ij} \overset{i.i.d.}{\sim} \mathcal{N}(0, \frac{\sigma^2}{n})$, then as $n$ grows large, the eigenvalues of the matrix become uniformly distributed inside the disk with radius $\sigma$ in the complex plane [51]. This result can be further generalized to more general random matrices where the i.i.d distribution is not Gaussian but has a finite variance.

**Theorem 1** *Let $A_n$ be the $n \times n$ random matrix whose entries are i.i.d. complex random variables with mean 0 and variance 1. The empirical spectral distribution of $1/\sqrt{n}$ then converges (both in probability and in the almost sure sense) to the uniform distribution on the unit disk [26].*

This theorem means that the circular law can not only be used in ideal Gaussian distribution but also can be used in the case of other distributions like sparse connection and biologically plausible log-normal distribution, etc. However, it still requires the distribution to be identical independent distribution, while there are multiple types of neurons and connection probabilities that decay with the distance between neurons. Therefore, we still can not use this theorem in biologically plausible spatially distributed neural networks.

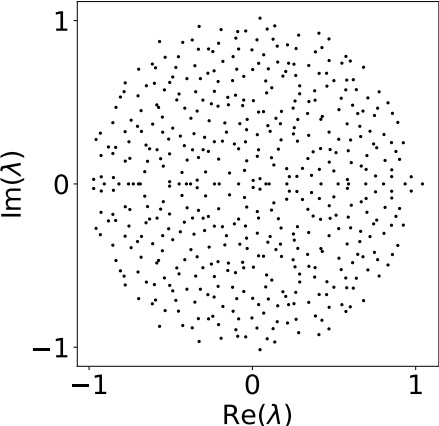

Figure 12: The eigenspectrum of a random matrix with elements from i.i.d Gaussian distribution with unit variance and zero mean.

Thanks to techniques of free probability theory [28], we can deal with the case that the distribution is independent, zero-mean but not identical. Besides, using dynamical mean-field theory can also derive similar results [29].

**Theorem 2** *Let $A_n = (\sigma_{ij})$ be an $n \times n$ deterministic matrix, and $X_n = (X_{ij})$ be an $n \times n$ random matrix with i.i.d. centered entries of unit variance. Define the rescaled matrix:*

$$Y_n = \frac{1}{\sqrt{n}} A_n \circ X_n,$$

*where $\circ$ denotes the Hadamard product. Let $\mu_n^Y$ denote the empirical spectral distribution (ESD) of $Y_n$. For variance profiles $\sigma_{ij}^2 = \sigma^2\left(\frac{i}{n}, \frac{j}{n}\right)$, $\mu_n$ has a positive density on the centered disc of radius $\sqrt{\rho(V_n)}$, where $V_n = \frac{1}{n}\sigma_{ij}^2$ and $\rho(V_n)$ is its spectral radius. [28]*

Using this theorem, we can deal with biologically plausible neural networks with multiple types of neurons and spatial distribution. Although the connection strength distribution between different types of neurons are different, and the distribution also varies with distance between neurons, we only need to calculate a profile matrix $\sigma_{ij}$ above to determine the radius of a eigenspectrum disk.

### A.13.2 Outliers of Eigenspectrum

The theorems mentioned above are all required of a zero-mean condition. However, in biologically plausible case, due to Dale's Law, the distribution of connection strength between certain types of neurons cannot have a zero-mean. For example, the connection strength between excitatory neurons must have a positive mean.

[43] considers a special case of non-zero-mean distributions. In their model, neurons are sparsely connected with a certain probability $p$ for both excitatory and inhibitory neurons. The connection strength are the same among excitatory synapses and inhibitory synapses separately. They showed that the eigenspectrum of the connectivity matrix is composed of a bulk disk part and an outlier. This outlier lies in the position of the mean connection strength of all synapses. Namely, the non-zero mean of a distribution creates outliers.

The example above is actually a special case of low-rank perturbation on a zero-mean random matrix. We can gain the following intuition. The connectivity matrix can be decomposed into two parts: a determined mean part and a zero-mean random matrix part. The zero-mean random matrix part creates a bulk disk eigenspectrum. And the determined mean part is actually a low-rank matrix because its rank is less than the number of types of neurons. This low-rank perturbation creates outliers in eigenspectrums [27] provided a rigorous mathematical theorem on outliers and low-rank perturbation.

**Theorem 3** *Let $X_n$ be an iid random matrix with finite fourth moment, and for each $n$, let $C_n$ be a deterministic matrix with rank $O(1)$ and operator norm $O(1)$. Assume that for large $n$, $C_n$ has no eigenvalues in $\{z \in \mathbb{C} : 1 + \varepsilon < |z| < 1 + 3\varepsilon\}$ and has $j = O(1)$ eigenvalues in $\{z \in \mathbb{C} : |z| \geq 1 + 3\varepsilon\}$. Then, almost surely, for large $n$, $\frac{1}{\sqrt{n}}X_n + C_n$ has exactly $j$ eigenvalues in $\{z \in \mathbb{C} : |z| \geq 1 + 2\varepsilon\}$, and these eigenvalues satisfy $\lambda_i\left(\frac{1}{\sqrt{n}}X_n + C_n\right) = \lambda_i(C_n) + o(1)$ as $n \to \infty$ for each $1 \leq i \leq j$ [27].*

With this theorem, we can finally characterize the eigenspectrum of biologically plausible neural networks. The sparsity and variance of connection strength both contributed to the zero-mean random part, which creates a bulk disk part of eigenspectrum. And the Dale's law force the connectivity matrix have a determined part. This part is often low-rank, thus creating outliers of the eigensepctrum.

### A.13.3 Eigenspectrum of connectivity matrices of spatially extended neural networks

As mentioned above, the eigenvalues of the connectivity matrix $\mathbf{J}$ consist of a circular bulk part and a set of outliers. [27] noted that for a random matrix subjected to a low-rank perturbation, the eigenvalues of the new matrix largely remain within the original circle, with any outliers located where the eigenvalues of the perturbation matrix lie. We can conceptualize the connectivity matrix $\mathbf{J}$ as comprising the expected values of each element, $\bar{\mathbf{J}}$, and the deviations from this expectation, $\delta\mathbf{J}$. The matrix $\delta\mathbf{J}$ corresponds to the aforementioned random matrix, while $\bar{\mathbf{J}}$ corresponds to the low-rank perturbation matrix. Therefore, $\delta\mathbf{J}$ dictates the bulk part of the eigenvalue spectrum, while $\bar{\mathbf{J}}$ determines the outlier part.

### A.13.4 Outliers

**Eigenvectors and the Spatial Translation Invariance of Neural Networks**   The matrix $\bar{\mathbf{J}}$ determines the outlier part of the eigenvalue spectrum [27]. In order to calculate the eigenvalues of the matrix $\bar{\mathbf{J}}$, we need to utilize its property of spatial translation invariance

Let's start with a relatively simple one-dimension case.

The expected component of the connectivity matrix, $\bar{J}$, can be expressed as a block matrix:

$$\overline{J} = \begin{bmatrix} \overline{J}^{EE} & \overline{J}^{EI} \\ \overline{J}^{IE} & \overline{J}^{II} \end{bmatrix},$$

where $\overline{J}^{\alpha\beta}$ represents the expected connectivity from neurons of type $\beta$ to neurons of type $\alpha$, forming a matrix of size $N_\alpha \times N_\beta$. It satisfies the following relation,

$$\overline{J}_{ij}^{\alpha\beta} = \frac{\overline{g}_{\alpha\beta}}{k_{\alpha\beta}^{out}} \cdot P_c^{\alpha\beta}(|x_i - x_j|).$$

The element $\overline{J}_{ij}^{\alpha\beta}$ depends only on the distance between neurons $i$ and $j$ and their respective types. In our model, the proportion of excitatory to inhibitory neurons satisfies $N_E : N_I = 4 : 1$. We can consider four excitatory neurons and one inhibitory neuron as forming a small unit, and the neural network consists of repeated instances of this unit. When the neural network is collectively translated by several unit distances in physical space, the matrix $\overline{J}$ remains unchanged, indicating its translational invariance.

We define the following block matrix $P$, which satisfies:

$$P = \begin{bmatrix} P^E & O \\ O & P^I \end{bmatrix},$$

where $P^\alpha$ is a matrix of size $N_\alpha \times N_\alpha$:

$$P_{ij}^E = \delta_{i,j-4}, \quad P_{ij}^I = \delta_{i,j-1}.$$

The matrix $P$ represents the translation of the neural network by one small unit and has the property:

$$P\overline{J}P^{-1} = \overline{J}.$$

This implies that the matrices $P$ and $\overline{J}$ share common eigenvectors.

The eigenvalues of $P$ satisfy $\lambda_n = e^{i\frac{k}{N_I}}$, where $k = 2\pi n, n = 0, \pm 1, \ldots, \pm\lfloor \frac{N_I}{2} \rfloor$. Each eigenvalue $\lambda_n$ is fivefold degenerate, with the eigenspace spanned by the vectors $\{\vec{u}_l^{(k)} | l = 0, 1, \ldots, 4\}$, which satisfy the following properties.

These vectors can be expressed as $u_l^{(k)} = [u_{El}^{(k)}, u_{Il}^{(k)}]^T$, where $u_{El}^{(k)}$ is a vector of length $N_E$ and $u_{Il}^{(k)}$ is a vector of length $N_I$.

For $l = 0, 1, \ldots, 3$:

$$[\vec{u}_{El}^{(k)}]_j = \delta_{l,j\,mod\,4} \cdot \frac{1}{\sqrt{N_I}} e^{i\frac{k}{N_I}\lfloor \frac{j}{4} \rfloor}, \quad [\vec{u}_{Il}^{(k)}]_j = 0.$$

For $l = 4$:

$$[u_{El}^{(k)}]_j = 0, \quad [u_{Il}^{(k)}]_j = \frac{1}{\sqrt{N_I}} \cdot e^{i\frac{kj}{N_I}}.$$

Thus, in the basis: $\{\vec{u}_l^{(k)} | l = 0, 1, \ldots, 4; \quad k = 2\pi n, \quad n = 0, 1, \ldots, N_I - 1\}$, the matrix $\overline{J}$ can be written as a block diagonal matrix, where each block corresponds to an effective connectivity matrix for a specific Fourier mode. The effective connectivity matrix for a Fourier mode with wave vector $k$ is given by:

$$\begin{aligned}
\left[\overline{J}^{(k)}\right]_{ll'} &= u_l^{(k)\dagger}\overline{J}u_{l'}^{(k)} = \sum_i \sum_j \left[u_l^{(k)}\right]_i \overline{J}_{ij}^{\alpha\beta} \left[u_{l'}^{(k)}\right]_j \\
&= \frac{1}{N_I} \sum_i \sum_j \exp(-ikx_i) \frac{\overline{g}_{\alpha\beta}}{k_{\alpha\beta}^{out}} \times \frac{k_{\alpha\beta}^{out}}{N_\alpha} \times \frac{1}{\sqrt{2\pi}d_{\alpha\beta}} \exp\left(-\frac{(x_i - x_j)^2}{2d_{\alpha\beta}^2}\right) \times \exp(ikx_j) \\
&\simeq \frac{1}{N_I} \int_0^1 dx \cdot N_I \int_0^1 dx' \cdot N_I \frac{\overline{g}_{\alpha\beta}}{N_\alpha} \times \frac{1}{N_\alpha} \exp\left(-\frac{(x_i - x_j)^2}{2d_{\alpha\beta}^2} + ik(x_i - x_j)\right) \quad (18) \\
&= \frac{N_I}{N_\alpha} \int_0^1 dx \frac{\overline{g}_{\alpha\beta}}{d_{\alpha\beta}} \exp\left(-\frac{x^2}{2d_{\alpha\beta}^2} + ikx\right) \\
&= \frac{N_I}{N_\alpha} \times \frac{\overline{g}_{\alpha\beta}}{d_{\alpha\beta}} \exp\left(-\frac{k^2 d_{\alpha\beta}^2}{2}\right).
\end{aligned}$$

This expression is accurate for long-wavelength modes. the matrix $\left[\overline{J}^{(k)}\right]$ takes the form:

$$
\overline{J}^{(k)} = \begin{bmatrix} \tilde{g}_{EE}^{(k)} & \cdots & \tilde{g}_{EE}^{(k)} & \tilde{g}_{EI}^{(k)} \\ \vdots & \ddots & \vdots & \vdots \\ \tilde{g}_{EE}^{(k)} & \cdots & \tilde{g}_{EE}^{(k)} & \tilde{g}_{EI}^{(k)} \\ \tilde{g}_{IE}^{(k)} & \cdots & \tilde{g}_{IE}^{(k)} & \tilde{g}_{II}^{(k)} \end{bmatrix}_{5\times5},
$$

where $\widetilde{g}_{\alpha\beta}^{(k)} = \frac{N_I}{N_\alpha}\overline{g}_{\alpha\beta} \cdot \exp\left(-\frac{d_{\alpha\beta}^2 \cdot k^2}{2}\right)$.

For the $k = 0$ Fourier mode, the outliers are precisely the two eigenvalues given by $\overline{J}_{eff}^{(K)}$, each of which is non-degenerate. For $k \neq 0$ modes, since the eigenvalues corresponding to $\pm k$ coincide, they exhibit twofold degeneracy.

For the two dimensions case, the mathematical derivation is almost the same. In the case of two dimensions, the spatial translation can be done in both directions. Therefore, the eigenvectors are plane waves.

**Effective Connectivity Matrix and Eigenvalues**    For each spatial Fourier mode, we can further simplify the effective connectivity matrix as follows:

$$
\mathbf{\overline{J}}_{\text{eff}}^{(k)} = \begin{bmatrix} \overline{g}_{EE} \exp\left(-\frac{||k||^2 d_{EE}^2}{2}\right) & \overline{g}_{IE} \exp\left(-\frac{||k||^2 d_{IE}^2}{2}\right) \\ \overline{g}_{EI} \exp\left(-\frac{||k||^2 d_{EI}^2}{2}\right) & \overline{g}_{II} \exp\left(-\frac{||k||^2 d_{II}^2}{2}\right) \end{bmatrix}. \tag{19}
$$

where $\vec{k} = (2\pi n_x, 2\pi n_y), \quad n_x, n_y = 0, \pm 1, \pm 2, \ldots$

As $k$ takes on different values, the effective connectivity matrices for different spatial Fourier modes yield different eigenvalues. These eigenvalues constitute the outliers in the eigenvalue spectrum of the connectivity matrix. In the case without spatial distribution, the number of outliers corresponds to the number of neuron types; in the presence of spatial distribution, as the spatial scale of the network increases, the number of outliers also increases and is ordered according to their corresponding wave vectors $k$. When the spatial scale approaches infinity, these outliers will be arranged along a continuous curve (here, we reference the figure from spatial effect).

The two-dimensional case is similar, except that the wave vector $k$ is a vector that takes values $\vec{k} = (2\pi n_x, 2\pi n_y), \quad n_x, n_y = 0, \pm 1, \pm 2, \ldots$. The eigenvalues in the two-dimensional case are also given by the effective connectivity matrix eigenvalues for different $k$ values. The eigenvectors corresponding to the outlier eigenvalues are provided by the spatial Fourier modes indicated by $k$.

### A.13.5   Bulk disk part

The matrix $\delta\mathbf{J}$ governs the circular part of the eigenvalue spectrum. [28] and [29] provide formulations for the eigenvalue distribution of random matrices with independent entries, mean zero, and finite variance. The eigenvalues are distributed within a circle of a specific radius $r = \sqrt{\max\left(\lambda(M)\right)}$, where the elements of the matrix $M$ represent the variances of the elements of the connectivity matrix $\mathbf{J}$. The variance of the elements of the connectivity matrix $\mathbf{J}$ is given by:

$$
M_{ij}^{\alpha\beta} = \mathbb{E}\left[\delta J_{ij}^{\alpha\beta2}\right] = p_c^{\alpha\beta}\left(|x_i - x_j|\right)\left(1 - p_c^{\alpha\beta}\left(|x_i - x_j|\right)\right) \times \frac{\overline{g}_{\alpha\beta}^2}{k_{\alpha\beta}^{out2}} + p_c^{\alpha\beta}\left(|x_i - x_j|\right)\frac{\sigma_{\alpha\beta}^2}{k_{\alpha\beta}^{out}}. \tag{20}
$$

Similar to the mean part, the matrix $M_{ij}^{\alpha\beta}$ is also a spatial translation invariant matrix, because both the variance and connection probability of the synapses between two neurons are only related to the relative distance between these two neurons. Therefore, the variance matrix $M$ can also be diagonalized into a series of small block matrices. Let's start with the one dimension case,

$$U^{-1}MU = \begin{bmatrix} M_{\alpha\beta}^{(k=0)} & 0 & 0 & \cdots & 0 \\ 0 & M_{\alpha\beta}^{(k=2\pi)} & 0 & \cdots & 0 \\ 0 & 0 & M_{\alpha\beta}^{(k=4\pi)} & \cdots & 0 \\ \vdots & \vdots & \vdots & \ddots & \vdots \\ 0 & 0 & 0 & \cdots & \end{bmatrix}.$$

Because the radius of the bulk disk of an eigenspectrum only depends on the largest eigenvalue of $M_{ij}^{\alpha\beta}$, and all the elements of $M_{ij}^{\alpha\beta}$ are positive, we only need to consider the zero wave vector submatrix $M_{\alpha\beta}^{(k=0)}$.

Zero wave vectors are spatially uniform, thus the elements of submatrix $M_{\alpha\beta}^{(k=0)}$ are the average of original elements of $M_{ij}^{\alpha\beta}$. For simplicity, we denote the matrix $M_{\alpha\beta}^{(k=0)}$ as $M_{\alpha\beta}$. s

For the one-dimensional case, this reduced matrix $M_{\alpha\beta}$ is:

$$\begin{aligned} M_{\alpha\beta} &= \sum_j \left( \delta J_{ij}^{\alpha\beta} \right)^2 \\ &= \int_0^{+\infty} dx \cdot N_\beta E \left[ \left( \delta J_{ij}^{\alpha\beta} \right)^2 \right] \\ &= \frac{N_\beta}{N_\alpha} \left[ \frac{\bar{g}_{\alpha\beta}^2}{k_{\alpha\beta}^{out}} \left( 1 - \frac{k_{\alpha\beta}^{out}}{2\sqrt{\pi} d_{\alpha\beta} \cdot N_\alpha} \right) + \sigma_{\alpha\beta}^2 \right]. \end{aligned} \tag{21}$$

For the two-dimensional case, the reduced matrix $M_{\alpha\beta}$ is:

$$\begin{aligned} M_{\alpha\beta} &= \sum_j \left( \delta J_{ij}^{\alpha\beta} \right)^2 \\ &= \int_0^{+\infty} dx \cdot 2\pi x \cdot N_\beta E \left[ \left( \delta J_{ij}^{\alpha\beta} \right)^2 \right] \\ &= \frac{N_\beta}{N_\alpha} \left[ \frac{\bar{g}_{\alpha\beta}^2}{k_{\alpha\beta}^{out}} \left( 1 - \frac{k_{\alpha\beta}^{out}}{4\pi d_{\alpha\beta}^2 \cdot N_\alpha} \right) + \sigma_{\alpha\beta}^2 \right]. \end{aligned} \tag{22}$$

### A.14 Phase Boundary of Chaos Phase

As mentioned above, the theoretical prediction based on linearization and eigenspectrum cannot fully characterize the dynamical behavior far away from a fixed point. The chaos phase is one of these cases. The bulk disk of an eigenspectrum corresponds to neural activity without spatial patterns, while outliers are related to spatially ordered neural activity. If both the radius of the bulk disk and the real part of outliers are larger than 1, they will compete against each other and we cannot directly determine whether the neural activity is spatially ordered. However, with the tool of dynamical mean-field theory (DMFT) [19], we can mathematically rigorously derive the condition for spatial patterns in the chaos phase.

For a neural network with the radius of a bulk disk part greater than 1, if the deviation of the mean of the neurons' membrane potential gradually amplifies after a perturbation, eventually the neural activities of different neurons will become synchronized, and the neural network will no longer be in the chaos phase. [52] calculated how the mean and variance of the neuronal membrane potential evolve after being subjected to perturbation,

$$(\mathbf{I} - \mathbf{D} + \partial_t)\chi^\Delta = \mathbf{E}(\bar{\mathbf{g}}\chi + \mathbf{I}\delta(t)), \tag{23}$$

$$(\mathbf{I} - \mathbf{A}\bar{\mathbf{g}} + \partial_t)\chi(t) = \mathbf{B}\chi^\Delta + \mathbf{A}\delta(t). \tag{24}$$

where,

$$A_{kl} = \delta_{kl}\langle\phi'_l\rangle + \langle\phi''_k\rangle\sigma^2_{kl}\langle\phi_l\phi'_l\rangle, \tag{25}$$

$$B_{kl} = \frac{1}{2}\sigma^2_{kl}\langle\phi''_k\rangle[\langle\phi'^2_l\rangle + \langle\phi_l\phi''_l\rangle], \tag{26}$$

$$D_{kl} = \sigma^2_{kl}[\langle\phi'^2_l\rangle + \langle\phi_l\phi''_l\rangle], \tag{27}$$

$$E_{kl} = 2\sigma^2_{kl}\langle\phi_l\phi'_l\rangle. \tag{28}$$

Here, taking the average $\langle\cdot\rangle$ refers to averaging over the distribution of the neuronal membrane potentials. The distribution of the neuronal membrane potentials follows a Gaussian distribution, the mean and variance of which can be theoretically calculated based on dynamic mean-field theory [52].

Let's denote the response matrix as follows,

$$\mathbf{R} = \begin{bmatrix} \mathbf{I} - \mathbf{A}\overline{\mathbf{g}} & -\mathbf{B} \\ -\mathbf{E}\overline{\mathbf{g}} & \mathbf{I} - \mathbf{D} \end{bmatrix}. \tag{29}$$

For a neural network with the radius of bulk disk part larger than 1, if all the eigenvalues of a response matrix are less than 0, the chaotic neural activity is stable and the neural activity is within a chaos phase.

We further extended this result to spatially extended neural networks. We can regard a neural network with spatial distribution to be composed of lots of populations with area $\Delta S$. Therefore, a neural network with spatial distribution is equivalent to a neural network with infinite populations.

In the model presented in the main text, the activation function for neurons is the hyperbolic tangent function, $\tanh$. The distribution of membrane potential is a Gaussian distribution with zero mean. Therefore, the matrix $\mathbf{B}$ and $\mathbf{E}$ are zero matrix. The response matrix $\mathbf{R}$ is a block diagonal matrix. The submatrix $\mathbf{I} - \mathbf{A}\overline{\mathbf{g}}$ in the upper left corner represents the mean of membrane potential response to external inputs, and the submatrix $\mathbf{I} - \mathbf{D}$ in the lower right corner represents the variance of membrane potential response to external inputs.

The stability of the mean of membrane potential determines the phase boundary between the chaos phase and other phases with locally synchronized neural activity. Therefore, the largest real part of eigenvalues of the matrix $\mathbf{A}\overline{\mathbf{g}}$ determines the phase boundary of the chaos phase.

Similar to the calculation above, the matrix $\mathbf{A}\overline{\mathbf{g}}$ is spatially translation invariant. Therefore, we can diagonalize this matrix into a series of submatrices corresponding to different wave vectors $k$. The elements of submatrices are as follows,

$$(\mathbf{A}\overline{\mathbf{g}})^{(k)}_{\text{eff}} = \begin{bmatrix} \langle\phi'_E(h^E_i)\rangle\overline{g}_{EE}\exp\left(-\frac{||k||^2 d^2_{EE}}{2}\right) & \langle\phi'_I(h^I_i)\rangle\overline{g}_{IE}\exp\left(-\frac{||k||^2 d^2_{IE}}{2}\right) \\ \langle\phi'_E(h^E_i)\rangle\overline{g}_{EI}\exp\left(-\frac{||k||^2 d^2_{EI}}{2}\right) & \langle\phi'_I(h^I_i)\rangle\overline{g}_{II}\exp\left(-\frac{||k||^2 d^2_{II}}{2}\right) \end{bmatrix}. \tag{30}$$

where $\vec{k} = (2\pi n_x, 2\pi n_y)$, $n_x, n_y = 0, \pm 1, \pm 2, \dots$. Averaging is performed over the distribution of neuronal membrane potentials. Using the tool of dynamical mean-field theory [52], the distribution of neuronal membrane potentials follows a Gaussian distribution. Let's denote the expectation of membrane potential as $u_\alpha$, the expectation of firing rate as $m_\alpha$, the variance of membrane potential as $\Delta_\alpha$, and the autocorrelation of firing rate as $C_\alpha$. They satisfy,

$$u_\alpha = \sum_l \overline{g}_{\alpha\beta}m_\beta + h^0_\beta, \tag{31}$$

$$m_\alpha = \langle\phi_\alpha(\sqrt{\Delta_\alpha}z + u_k)\rangle, \tag{32}$$

$$\Delta_\alpha = \sum_\beta \frac{N_\beta}{N_\alpha}\left[\frac{\overline{g}^2_{\alpha\beta}}{k^{out}_{\alpha\beta}}\left(1 - \frac{k^{out}_{\alpha\beta}}{4\pi d_{\alpha\beta}^2 \cdot N_\alpha}\right) + \sigma_{\alpha\beta}^2\right]C_\beta + \xi^2_0, \tag{33}$$

$$C_\alpha = \langle\phi^2(\sqrt{\Delta_\alpha}z + u_\alpha)\rangle. \tag{34}$$

where $h_\beta$ is the direct current (DC) input to the $\beta$ type of neurons. $z$ follows a standard Gaussian distribution with unit variance.

If the eigenvalues of all the submatrices are less than 1, the chaotic neural activity is stable and the neural network is in the chaos phase. Otherwise, the chaotic neural activity is unstable. We can assign its phase based on the wave vector $k$ and whether the eigenvalues of instability are complex to determine which phase it belongs to.

