# OpenReview forum: "Bridging Scales: Spectral Theory Reveals How Local Connectivity Rules Sculpt Global Neural Dynamics in Spatially Extended Networks"
_NeurIPS.cc/2025/Conference — NeurIPS 2025 poster_

### Official Review · Reviewer_eB7W · 2025-06-28

**Clarity:** 3
**Significance:** 3
**Originality:** 3
**Rating:** 5
**Confidence:** 3

**Summary:**

This paper investigates the underlying mechanisms through which macroscopic dynamics arise from microscopic neural circuitry. By integrating a spatially extended neural network with spectral theory, the authors propose a novel framework, which is substantiated by numerical experiments.

**Questions:**

- What does $J$ represent in Equation (1)? Does it refer to the elements of the connection matrix?
- Are there any other related works in this research area? If so, how does this paper differ from or improve upon those existing studies?

**Ethical Concerns:**

["NO or VERY MINOR ethics concerns only"]

**Final Justification:**

This paper tackles a research problem that is both significant and novel. The presented numerical simulations are sound and effectively validate the proposed method. All of my previous concerns have been satisfactorily addressed, and I have no further comments.

**Limitations:**

See Weaknesses and Questions.

**Quality:**

3

**Strengths And Weaknesses:**

**Strengths**
- The research problem addressed in the paper is both important and novel.
- The numerical simulation results appear to be reasonable and support the proposed approach.

**Weaknesses**
- Some notations are not clearly defined. For instance, in line 77, it is unclear what the symbol ‘[.]’ denotes—does it represent a rounding function?
- While the use of the activation function $\tanh(x)$ is standard, the appearance of the expression $5\tanh(x/5)$ is unusual and lacks sufficient explanation or justification.

---

> ### Author Rebuttal · Authors · 2025-07-30
>
> Dear Reviewer,
>
> Thank you very much for your careful reading and for the helpful feedback.  We appreciate the positive assessment of the novelty, significance, and technical soundness of our work, as well as the specific suggestions for improving clarity.  Below we address every point that you raised.
>
> **W1: Some notations are not clearly defined. For instance, in line 77, it is unclear what the symbol ‘[.]’ denotes—does it represent a rounding function?” (Line 77)**
>
> Thank you for point out that the symbol here lack of clarity. The notation $
> [\cdot]_+$ means,
>
> $$
> [\cdot]_+ = \begin{cases}
> 0 & \text{if } x \leq 0, \newline
> x & \text{if } x > 0.
> \end{cases}
> $$
>
> This symbol is common in the computational-neuroscience literature(for example, Kadmon & Sompolinsky, 2015), yet it still requires further clarification. In the final version, we will revise this notation to $ReLU(x)$
>
> **W2: While the use of the activation function $\tanh(x)$ is standard, the appearance of the expression $5\tanh(x/5)$ is unusual and lacks sufficient explanation or justification.**
>
> We use the activaton function of inhibitory neurons $5\tanh(x/5)$ for its biological plausibility. The saturated firing rate of inhibitory neurons is substantially higher than that of excitatory neurons (Wang B et al., 2016; Zhong P et al., 2011; Sanzeni A et al., 2020). The activation function of inhibitory neurons is $5\tanh(x/5)$ while that of excitatory neurons is $\tanh(x)$, so that the saturated firing rate of inhibitory neurons is higher than that of excitatory neurons, which is consistent with experimental evidence. In the final version, we will add the explanation of activation functions.
>
> **Q1: What does J represent in Equation (1)? Does it refer to the elements of the connection matrix?**
>
> Yes, it refer to the elements of the connection matrix. To be specific, $J^{\alpha\beta}_{ij}$ weight from neuron j to neuron i (positive for excitatory, negative for inhibitory), and neuron i belongs to type $\alpha$ and neuron j belongs to type $\beta$. We will further clarify the meaning of $J$ in the final version.
>
> **Q2: Are there any other related works in this research area? If so, how does this paper differ from or improve upon those existing studies?**
>
> Thank you for raising the question about related work.  We apologize for the oversight: a full comparison with existing literature is already provided in Appendix A.1, but we failed to direct the reader to it from the main text.
>
> In the revised manuscript we will:
>
> Add a forward citation in Section 1 (Introduction) that reads:
> “For a detailed comparison with prior approaches, including neural field theory, numerical simulation and random matrix theory, please refer to Appendix A.1."
>
> We appreciate your patience and the helpful pointer.
>
>
> ### reference ###
> Kadmon J, Sompolinsky H. Transition to chaos in random neuronal networks[J]. Physical Review X, 2015, 5(4): 041030.
>
> Wang B, Ke W, Guang J, et al. Firing frequency maxima of fast-spiking neurons in human, monkey, and mouse neocortex[J]. Frontiers in cellular neuroscience, 2016, 10: 239.
>
> Zhong P, Yan Z. Differential regulation of the excitability of prefrontal cortical fast-spiking interneurons and pyramidal neurons by serotonin and fluoxetine[J]. PloS one, 2011, 6(2): e16970.
>
> Sanzeni A, Akitake B, Goldbach H C, et al. Inhibition stabilization is a widespread property of cortical networks[J]. elife, 2020, 9: e54875.

---

> > ### Comment · Reviewer_eB7W · 2025-08-01
> > **The authors have addressed all my concerns.**
> >
> > The authors have addressed all my concerns and  I have no more questions.

---

### Official Review · Reviewer_wB1x · 2025-07-01

**Clarity:** 2
**Significance:** 3
**Originality:** 3
**Rating:** 5
**Confidence:** 3

**Summary:**

This paper introduces a theoretical framework linking the microscopic architecture of spatially extended neural networks to their emergent macroscopic dynamics. Using a rate-based model of excitatory and inhibitory neurons on a 2D grid with distance-dependent connectivity, the authors develop a spectral theory, grounded in random matrix theory, for the network’s effective connectivity. They show that local parameters shape the eigenvalue spectrum, comprising a bulk disk and distinct outliers tied to spatial Fourier modes. This spectrum explains a diverse range of dynamics, such as synchrony, oscillations, bumps, waves, and chaos, within a unified model, and aligns qualitatively with cortical dynamics observed in mice.

**Questions:**

- The paper introduces the key concept of local sparsity, which plays a central role in determining the bulk radius and chaos onset. Could you clarify how this quantitatively refines the standard notion of sparsity in non-spatial random networks?

- The SVD analysis of the phase velocity field (Fig. 4, A.2) offers a strong link to experimental data. Could a more direct quantitative connection be made between the dominant SVD modes and the leading eigenvectors of the theoretical effective connectivity matrix?

-The model assumes isotropic connectivity. How might the spectral theory extend to anisotropic connectivity?

**Ethical Concerns:**

["NO or VERY MINOR ethics concerns only"]

**Final Justification:**

I appreciate the authors’ initial rebuttal and clarifications. As noted in my original review, the paper presents a rigorous theoretical framework grounded in random matrix theory, effectively applied to spatially extended networks, and supported by robust analytical and numerical evidence, with extensive supplementary material reinforcing its quality. My major concerns regarding anisotropic extensions and the influence of outliers were properly addressed.

Accordingly, I maintain my original rating: 5 – Accept.

**Limitations:**

Yes

**Quality:**

3

**Strengths And Weaknesses:**

Strengths:

The theoretical framework is built upon solid foundations in random matrix theory and is meticulously applied to spatially extended networks. The claims are well-supported by a combination of analytical derivations and extensive numerical simulations. The appendix provides a wealth of supporting material that confirms the paper's quality.

Weaknesses:

The linearization around a fixed point is a necessary simplification for the initial theory. While Appendix A.8 acknowledges this limitation and Appendix A.11 employs DMFT to address it near the chaos boundary, a full theoretical account of dynamics far from the fixed point, such as large-amplitude limit cycles, remains an open challenge, as expected for such complex systems. Some sections of the paper are difficult to follow due to limited context and abrupt transitions. Providing clearer background and improving the narrative flow would significantly enhance readability.

---

> ### Author Rebuttal · Authors · 2025-07-30
>
> Dear Reviewer,
>
> Thank you for your careful reading and constructive feedback. Below we address each of your specific comments in detail.
>
> **W1: Some sections of the paper are difficult to follow due to limited context and abrupt transitions. Providing clearer background and improving the narrative flow would significantly enhance readability.**
>
> Thank you for highlighting the issue of narrative flow. We agree that the original presentation placed the final theoretical statements in the main text while relegating all detailed derivations to the appendices without adequate sign-posting, making the exposition difficult to follow.
>
> To remedy this, we will implement the following changes:
>
> 1. Section 1 (Introduction) will now include a concise “Road-map” paragraph as follows,
>
> "We first define the spatial E/I network (Sec 2), then analytically derive its bulk-plus-outlier spectrum (Sec 3.2, App A.10). The spectrum predicts six dynamical phases (Sec 3.3, App A.8 & A.11). Phase transitions are studied via SVD of phase-velocity fields (Sec 3.5, App A.2). Parameters and numerical validation are in App A.4–A.7."
>
> 2. Provide more specific references to the appendix. For example, in Section 3.2.2 “Bulk Disk,” we will add explicit references to the specific theorems and equations in A.10 so that the derivation is clearer.
>
> We believe these two revisions will improve readability and coherence.
>
> **Q1: The paper introduces the key concept of local sparsity, which plays a central role in determining the bulk radius and chaos onset. Could you clarify how this quantitatively refines the standard notion of sparsity in non-spatial random networks?**
>
> For clarity, the key differences are summarized below.
>
> | Aspect                   | Standard sparsity                                                                                                         | Local sparsity                                                                                                            |
> | ------------------------ | ------------------------------------------------------------------------------------------------------------------------- | ------------------------------------------------------------------------------------------------------------------------- |
> | Expression               | $\frac{k^{out}_{\alpha\beta}}{N_{\alpha}}$                                                                  | $\frac{k^{out}_{\alpha\beta}}{\pi d_{\alpha\beta}^{2}\,N_{\alpha}}$                                         |
> | Interpretation           | Fraction of connections a neuron makes relative to the **total** number of neurons                                        | Fraction of connections relative to the expected number of neurons **within its axonal spread** $\pi d_{\alpha\beta}^{2}$ |
> | Scaling with system size | Decreases as $1/N_{\alpha}$ when the tissue area grows while $k^{out}_{\alpha\beta}$ is kept fixed | Remains constant, because number of neurons within axonal spread $\pi d_{\alpha\beta}^{2}$ remains constant.                          |
>
>
> Therefore, unlike standard sparsity, local sparsity remains invariant across spatially extended neural networks of different sizes that share identical structural properties, providing a better characterization of network sparsity and prediction of its dynamical behavior.
>
> In the revised manuscript, we will explicitly present the expression for standard sparsity and clearly delineate its differences from local sparsity to enhance clarity.
>
> **Q2: Could a more direct quantitative connection be made between the dominant SVD modes and the leading eigenvectors of the theoretical effective connectivity matrix?**
>
> Thank you for kindly suggesting that we could find a connection between the dominant SVD modes and the leading eigenvectors of the connectivity matrix. If I didn't misunderstand you, the SVD modes are vector fields while the leading eigenvectors are scalar fields, which are hard to compare. Instead, we have established a directly quantitative connection between scalar fields of neural activity of model and experiment, which will be added to the revised manuscript after section 3.5.
>
> To be specific, we have used order parameters to quantitatively characterize the similarity of simulation and experimental neural activity, finding a preliminary corresponding relationship between different degrees of consciousness and phases in our model.
>
> Firstly, by utilizing the order parameters of mean activity, local synchronization, oscillation index, as is used in the paper, we can exclude the possibility of asynchrony, synchrony, chaos, bump phase, and tell that the experimental states of different degree of consciousness can only be in either the oscillation phase or the wave phase. When distinguishing between these two phases, we didn't use Moran's index as in the paper because we found that the essential difference between unconscious and conscious states is the degree of localization of patterns, which can't be characterized by Moran's index. Instead, we used persistent homology, which is used to characterize the degree of spatial localization in a 2D scalar field $f(x, y)$. Larger persistent homology means a higher degree of pattern localization.
>
> By comparing the persistent homology of the experimental activity and the simulated activity, we can quantify their degree of similarity and find the corresponding phase of different degrees of consciousness.
>
> In addition, we also tried two other methods to find a quantitative connection between theory and experiment, but they all have limitations explained below.
>
> 1. We tried to compare the SVD modes of simulation and experiment data (Fig.4 (c)(d)). However, we found it tricky because our model does not have a typical scale like the brain. If we want to compare them, we have to consider scaling of the vector fields, but we don't know how much to scale. For example, a sink-pattern vector field will look like plane wave at some distance away from the singularity if it is enlarged to a certain extent, which will lead to fallacies when being compared to the experimental SVD modes.
>
> 2. We tried to compare the leading eigenvectors of the connectivity matrix and the leading spatial mode of dynamical mode decomposition (DMD) of the experimental data. DMD can decompose the activity $f(x,y,t)$ into the following linear combination form:
>     $$f(x,y,t) = \sum_{i=1}^{n} b_i \phi_i(x,y) \exp(\omega_i t),$$
>     where $b_i$ is the amplitude of DMD mode $i$, and $\phi_i(x,y)$ is the spatial part of the DMD mode.
>
>     We used persistent homology to compare the DMD spatial modes and the leading eigenvectors. However, we found that the leading DMD spatial mode can not well reflect the pattern localization properties of the experimental activity, sometimes matching the property while sometimes not, depending on the specific method of DMD.
>
> **Q3: The model assumes isotropic connectivity. How might the spectral theory extend to anisotropic connectivity?**
>
> We now provide three possible generalizations of anisotropic connectivity. All derivations assume the connection probability (Eq. 2) is modified while the spatial translation invariance of the mean connectivity is retained.
>
> 1. Shift of the projection range center
>
>    we can introduce anisotropy by shifting of the projection range center, which means that,
>
>    $$ p_c^{\alpha\beta}(x_i - x_j; y_i - y_j) = \frac{k^{out}_{\alpha\beta}}{N_{\alpha}} g(x_i - x_j + \Delta x; d_{\alpha\beta}) g(y_i - y_j + \Delta y; d_{\alpha\beta})$$
>
>     In this case, the connection probability of one neuron projecting to another located at an offset $(\Delta x, \Delta y)$ is maxima.
>
>     **a. Bulk disk part**
>
>     The bulk disk part would remain unchanged under this kind of anisotropy.
>
>     According to Equation 4. and Appendix A.10, the radius of bulk disk part only depends on the variance matrix M. The shift of the projection range center only results cyclic row permutation of variance matrix M, which leaves all eigenvalues of matrix M—and hence the bulk radius—unchanged.
>
>     **b. Outliers**
>
>     Currently, the outliers no longer lie along curves or lines, but instead spread over a certain region of the complex plane.
>
>     According to Equation 3 and Appendix A.10.4, each outlier has a specific wave-vector $\vec{k} = \left( 2\pi n_x, 2\pi n_y \right), \quad n_x, n_y = 0, \pm 1, \pm 2, \cdots$. In the isotropic case, outliers only denpend on the norm of wave-vector $\Vert k \Vert$, which is "one-dimentional", therefore outliers lie along curves or lines. However, in the anisotropic case, outliers depend on $(k_x, k_y)$, and spread over a certain region of the complex plane.
>
>     The calculation of outliers in the anisotropic case is similar to Appendix A.10.4. Because the expectation part of connection matrix is still spatially-translation invariant, we can still obatain its eigenvectors and eigenvalues following the same procedure of Appendix A.10.4.
>
> 2. projection-range anisotropy
>
>     Another way to introduce anisotropy is to have different projection range at different axis.
>
>     $$p_c^{\alpha\beta}(x_i - x_j; y_i - y_j) = \frac{k^{out}_{\alpha\beta}}{N_{\alpha}} g(x_i - x_j; d^x_{\alpha\beta}) g(y_i - y_j; d^y_{\alpha\beta})$$
>
>     **a. Bulk disk part**
>
>     The radius of bulk disk part only depends on the variance matrix $M^{\alpha\beta}_{ij}$. Therefore, we can follow the same procedure of Appendix A.5.10 to calcualte it.
>
>     **b. Outliers**
>
>     As in case 1, the outliers fill a two-dimensional region labelled by $(k_x​,k_y​)$; their exact positions follow from the same Fourier analysis detailed in Appendix A.10.4.
>
>
> We appreciate your patience and helpful suggestions.

---

> > ### Comment · Reviewer_wB1x · 2025-08-06
> > **Authors effectively addressed my comments.**
> >
> > I appreciate the authors thorough responses to the feedback provided. In my opinion, the manuscript merits acceptance.

---

### Official Review · Reviewer_yuaB · 2025-07-03

**Clarity:** 3
**Significance:** 3
**Originality:** 1
**Rating:** 4
**Confidence:** 3

**Summary:**

The paper studies a spatially extended neural network model with the goal of linking neural structure to dynamics. It applies spectral theory of random matrices to the analysis of its connectivity matrix. Specifically outlier eigenvalues can be interpreted in the current case. As in earlier work, asynchronous states, synchronous states, oscillations, localized activity bumps, traveling waves are identified, but although the comparison to cortical dynamics is attempted, qualitative comparisons remain difficult, while qualitative similarities are observable in some of the theoretically predicted cases as discussed earlier by (the cited paper by) Ermentrout.

**Questions:**

Considering that consciousness was used as a motivation in the abstract and introduction and a relevant paper is cited in the appendix, but is is not further discussed in the main paper, can any conclusions regarding consciousness be drawn from the results shown here?

Likewise potential biomarkers for neurological disorders are mentioned. Can you explain how the biomarkers could possibly be related to your results?

Can you comment on the similarity of the obtained patterns and the measurements in neuroscience? It seems there are some similarities which, however, leave other interpretations open. It will be necessary to avoid overstating the explanatory power of the current level of modeling.

Can you add results on the mode coupling that are necessary to understand the formation of the obsevable patterns? There are many examples in the literature related to physics systems but also in the context of neural systems that you can follow. In prinicple, it should be possible to obtain the phase diagrams theoretically, although there are some hysteretic effects that are not easy to obtain, but which have also here not been studied numerically.

**Ethical Concerns:**

["NO or VERY MINOR ethics concerns only"]

**Final Justification:**

In order for the paper to be accepted, the speculations on consciousness, biomarkers for neurological disorder, or understanding brain functions need to be more reasonably formulated. Although the rebuttal discussed certain limitations in these respects, this needs to be conveyed to the reader of the paper as well in unambiguous statements. In addition, the "unique sales point" over papers by Shunichi Amari, Haim Sompolinski, Gregor Shcoener and others needs to be made more clear. The phenomena described here are well known in the literature, but here the argument does indeed "stops short of drawing a solid conclusion" regarding the significance of these phenomena for brain function although the reader would expect this from the abstract.

**Limitations:**

Limitations are largely will described, but some points are raised in the questions above.

**Paper Formatting Concerns:**

References pertaining only to the appendix should be listed only after the appendix.

**Quality:**

3

**Strengths And Weaknesses:**

Much is known about the spectral theory of related systems, see the classical work of Swift and Hohenberg or (in the context of neural systems) of Amari in the 1970s, or Brunel and others in later years, which implies that the study of spectra of spatially extended systems based on superposition of Fourier modes is meanwhile a standard procedure. In order to understand relations between stripe-like and hexagonal patters, a second-order stability analysis is needed which is barely started in eq. 18 in the appendix, but it is not developed prodcutively, so that the main results of the paper are numerical although in a good correspondence with the RMT results. It is still a pity that the 20th century standard is not reached in this otherwise well-research study.

A main criticism is the interpretation of results in terms of biology and neuropsychology, see the questions below.

Phase diagrams like in Fig 2a and Fig. 4A have been shown frequently, so that it is not clear what exactly is the new contribution in the present case.

---

> ### Author Rebuttal · Authors · 2025-07-30
>
> Dear Reviewer,
>
> We sincerely appreciate the time and care you have devoted to evaluating our manuscript. Your incisive questions and constructive suggestions have helped us recognize where the presentation can be sharpened and where additional analysis will strengthen the narrative. Below, we address each point in detail.
>
> **W1:Phase diagrams like in Fig 2a and Fig. 4A have been shown frequently, so that it is not clear what exactly is the new contribution in the present case.**
>
> We thank the reviewer for pointing out that the role of the phase diagrams (Fig. 2a, 4a) was not sufficiently clear. These panels illustrate how connectivity features (e.g., E/I balance, E-I loop, projection range and so on) determines dynamics and are the direct visual counterparts of Section 3.4. Specifically:
>
> • Fig. 2a: x-axis = excitation–inhibition balance; y-axis = excitation–inhibition loop; comparision between two panels: mismatch of Excitation/Inhibition projection range.
>
> • Fig. 2b: shows how connection-weight variance drives chaos.
>
> We will add explicit captions and text links to make these relationships transparent in the revised manuscript.
>
> **Q1: "Considering that consciousness was used as a motivation in the abstract and introduction and a relevant paper is cited in the appendix, but is is not further discussed in the main paper, can any conclusions regarding consciousness be drawn from the results shown here?"**
>
> Thank you for this important question. We hope our framework can draw conclusions regarding consciousness as follows:
>
> 1. We hope to learn which dynamical phase corresponds to conscious versus unconscious states.  As sketched in Figure 3(b), each phase is distinguished by a unique set of order parameters; by estimating these parameters from empirical recordings, we hope to label the brain’s state as “conscious” or “unconscious” in a phase-specific manner.
> 2. With the random-matrix tools introduced, we can already predict how the spectrum of the effective connectivity matrix J selects a given phase. Besides, we discuss how specific connectivity characteristics select phases in section 3.4. We therefore hope to invert this mapping: once we know which phase is linked to consciousness, we hope to identify the characteristics of the connection structure (such as EI balance, EI loop, projection range) that place the system in that phase.
> 3. Ultimately, we hope to isolate the component of effective connections whose preservation keeps the brain in the “conscious” phase and whose disruption pushes it into an unconscious one.
>
> We tried to draw the above conclusions but they were not shown in the paper because we lacked appropriate methods to compare simulation and experiment. But now we have found a way to establish a preliminary relationship between consciousness degree and the corresponding phase in our model, which is explained below and will be added to in the revised manuscript after section 3.5.
>
> 1. By utilizing the order parameters of mean activity, local synchronization, oscillation index, as is used in the paper, we can exclude the possibility of asynchrony, synchrony, chaos, bump phase, and tell that the experimental states of different degree of consciousness can only be in either the oscillation phase or the wave phase.
>
> 2. When distinguishing between oscillation and wave, we didn't use Moran's index as in the paper because we found that the essential difference between unconscious and conscious states is the degree of localization of patterns, which can't be characterized by Moran's index. Instead, we used persistent homology (see *PS* for detailed definition). It can quantify the degree of pattern localization and distinguish between the oscillation and wave phase at the same time.
>
> 3. By comparing the persistent homology of the experimental activity and the simulated activity, we can know the corresponding phase of different degrees of consciousness.
>
> However, the current work stops short of drawing a solid conclusion for the reason of absence of solid evidence for a Unique-Phase brain: We currently lack solid evidence that, at any given moment, the brain resides in a single, well-defined phase. Instead, multiple phase patterns may superimpose or coexist simultaneously. Numerically, we have yet to devise suitable order parameters to capture such coexistence. Theoretically, by neglecting the mode-coupling effects you rightly highlighted (See Q4), we were unable to predict scenarios in which two phases coexist or superimpose. Consequently, although consciousness served as our motivation, we have not been able to discuss it in the depth it deserves. We hope to fill this gap in our future works.
>
> *PS*: Definition of persistent homology. Persistent homology is used to characterize the degree of spatial localization in a 2D scalar field $f(x, y)$. The method is to set a threshold $t$ and increase it from the minimum to the maximum value of $f(x, y)$, and then recognize connected components in the sublevel sets $X_t=${$(x,y)|f(x,y) \le t$} at each threshold value. As we sweep through values of $t$, new connected components (blobs) are born, and existing ones merge or vanish. Each such event is recorded as a birth-death pair $(b_i, d_i)$, representing the threshold at which a component appears and disappears respectively. The lifetime of a blob is defined as $d_i-b_i$. The persistent homology of $f(x, y)$ is defined as the sum of the lifetime of all the blobs. Larger persistent homology means a higher degree of pattern localization. The advantage of persistent homology is that it is parameter-free and scale-invariant.
>
> **Q2: "Likewise potential biomarkers for neurological disorders are mentioned. Can you explain how the biomarkers could possibly be related to your results?"**
>
> Different phases in our model have different corresponding effective connectivity (3.1 & 3.4 in paper).
> We have found a preliminary relationship between degrees of consciousness and the phases (see in answer to Q1), from which we can establish a connection between effective connectivity and conscious states. This may have the potential to be used as a biomarker for neurological disorders.
>
> **Q3: "Can you comment on the similarity of the obtained patterns and the measurements in neuroscience? It seems there are some similarities which, however, leave other interpretations open. It will be necessary to avoid overstating the explanatory power of the current level of modeling."**
>
> As is discussed in the answer to Q1, we can use persistent homology to quantitatively characterize the similarity of patterns in our simulation and in experimental data of different degrees of consciousness. However, we indeed recognize that these similarities do not provide a definitive explanation and that other interpretations are possible. We will take care to avoid overstating the explanatory power of our current level of modeling.
>
> **Q4: "Can you add results on the mode coupling that are necessary to understand the formation of the obsevable patterns?"**
>
> We thank the reviewer for the perceptive comments. The suggestion to employ mode-coupling analysis indeed offers a principled route to (i) identify sub-phases within one phase, (ii) predict the phase boundary when several eigenvalue outliers cross the instability threshold simultaneously. Below we outline how we will incorporate these ideas in the revised manuscript and in forthcoming work.
>
> 1. Degenerate-mode coupling and sub-phase selection
>
>     Numerical integration shows that the “bump phase” may adopt either stripe or hexagonal spatial profiles. At present we label all stripe/hexagon states as “bump”. A weakly nonlinear expansion around the homogeneous fixed point will allow us to derive coupled Landau equations for the slowly varying amplitudes $A_k$ of the marginally stable modes k with $|k| = k_c$.
>
>     For the Swift–Hohenberg class, the quadratic coefficient $\alpha_2$ in the Taylor expansion of the activation function determines the pattern selection: $\alpha_2 = 0$ selects stripes, whereas $\alpha_2 \neq 0$ selects hexagons via a resonant triad $k_1 + k_2 + k_3 = 0$. In our main text the tanh non-linearity has $\alpha_2 = 0$ and stripe-like neural activity patterns, while the quadratic non-linearity used in Appendix A.5 has $\alpha_2 \neq 0$ and has hexagonal neural activity patterns, consistent with this principle. We hope to perform more detailed calculation in future works.
>
>     How to carry out an analogous derivation within the RMT framework remains an open question. Due to a full derivation is beyond the scope of the present revision, we hope to address it in future work.
>
> 2. Coupling between outliers of distinct phases
>
>    When multiple eigenvalue outliers satisfy $Re (\lambda_i) > 1$, our current criterion—assigning the phase associated with the largest $Re(\lambda_i)$—works empirically but is not derived from first principles. By expanding the tanh activation function up to third order and tracking the time evolution of the amplitudes of two modes, we obtain the following equations,
>
>     $$\begin{cases}
>     \partial_t A_1 = \mu_1 A_1 - g_{1111} |A_1|^2 A_1 - g_{1112} |A_1|^2 A_2 - g_{1122} |A_2|^2 A_1 - g_{1222} |A_2|^2 A_2, \newline
>     \partial_t A_2 = \mu_2 A_2 - g_{2222} |A_2|^2 A_2 - g_{2221} |A_2|^2 A_1  - g_{2112} |A_1|^2 A_2 - g_{2111} |A_1|^2 A_1.
>     \end{cases}$$
>
>     with $\mu_i = \lambda_i − 1$, predicts either (i) one-mode-dominant, (ii) mixed patterns, or (iii) winner-take-all dynamics, depending on the signs and magnitudes of the cross-coupling coefficients $g_{ijkl}$. These coefficients can be computed explicitly from connection matrix parameters. Thus, this approach allows us to obtain the phase boundary when several eigenvalue outliers simultaneously cross the instability threshold.
>
>     We appreciate the reviewer’s guidance. As a full derivation would require substantial space, we defer it to a future study, where we hope to explore it in greater depth.

---

> ### Comment · Reviewer_yuaB · 2025-08-06
>
> In order for the paper to be accepted, the speculations on consciousness, biomarkers for neurological disorder, or understanding brain functions need to be more reasonably formulated. Although the rebuttal discussed certain limitations in these respects, this needs to be conveyed to the reader of the paper as well in unambiguous statements. In addition, the "unique sales point" over papers by Shunichi Amari, Haim Sompolinski, Gregor Shcoener and others needs to be made more clear. The phenomena described here are well known in the literature, but here the argument does indeed "stops short of drawing a solid conclusion" regarding the significance of these phenomena for brain function although the reader would expect this from the abstract.

---

> ### Author Response · Authors · 2025-08-08
> **Our revisions in the paper concerning the problems and suggestions mentioned in the above comment by reviewer yuaB**
>
> 1. **"Unique sales point" over papers by Shunichi Amari, Haim Sompolinsky, Gregor Schöner, and others**
>
>     We thank the reviewer for pointing out the need to better clarify the unique contribution of our work. A new subsection has been added to the **Related Works** section accordingly. The main points are summarized below:
>
>     1.1. **Relation to classical neural field theory**
>     We will further clarify that phenomena such as traveling waves, bump states, and lateral inhibition—studied in our manuscript—have been well documented in classical neural field theory by Amari, Schöner, Ermentrout, and others. These aspects are consistent with prior work and are not our primary novelty.
>
>     1.2. **Our novel contribution**
>     Our main contribution is a new framework based on **random matrix theory** for analyzing the dynamics of spatially extended neural networks. Unlike classical neural field models, our approach does **not assume homogeneous connectivity or the continuum limit**, allowing us to uncover **chaotic dynamics** absent in traditional neural field theory.
>     While such chaotic regimes have been thoroughly studied in non-spatial networks via dynamical mean field theory (DMFT) (e.g., Sompolinsky, Brunel), they remain underexplored in spatially structured settings. Our RMT-based analysis bridges this gap, and reveals a unifying and elegant perspective that **outliers** correspond to **Fourier modes (as in neural field theory)** and the **bulk spectrum** reflects **DMFT-like statistics**—a connection that may not be apparent without the RMT framework.
>
>     1.3. **Technical contribution**
>     As detailed in Section A.1 Supplementary Discussion, we will explain how our approach differs and offers new tools and insights into the structure-dynamics relationship in spatially extended networks compared to other works that also analyze neural dynamics via eigenspectrum methods.
>
> 2. **"...although the reader would expect this from the abstract."**
>
>     To avoid any possible over-statement, we have revised the abstract as follows,
>
>     2.1. (line 3) "We address this by developing a ..."\
>     **Revision**: "We take a step in this direction by developing a ..."
>
>     2.2 (line 12) "our framework not only explains ..."\
>     **Revision**: "our framework not only provides a possible explanation for"
>
>     2.3 (line 13) "...also offers a principled approach to inferring underlying effective connectivity changes from macroscopic brain activity."\
>     **Revision**: "...also offers a principled starting point for inferring underlying effective connectivity changes from macroscopic brain activity."
>
>     2.4 (line 14) "By mechanistically linking neural structure to dynamics, this work provides a powerful tool for understanding brain function and paves the way for identifying potential biomarkers for neurological disorders."\
>     **Revision**: "By mechanistically linking neural structure to dynamics, this work advances a principled framework for dissecting how large-scale activity patterns—central to cognition and open questions in consciousness research—arise from, and constrain, local circuitry."
>
> 3. **In order for the paper to be accepted, the speculations on consciousness, biomarkers for neurological disorder, or understanding brain functions need to be more reasonably formulated.**
>
>     To address the concern that speculations on consciousness and biomarkers were not reasonably formulated, in addition to the changes already mentioned in our previous rebuttal, we have made the following further revisions,
>
>     3.1. (line 50) "Our work thus provides a principled understanding of how network structure dictates emergent function."\
>     **Revision**: "Our work thus provides a principled understanding of how network structure dictates emergent neural activity."
>
>     3.2. (line 246) "Thus, the identified dynamical phases are attractor states within a larger phase space through which the brain navigates, enabling cognitive flexibility."\
>     **Revision**: "Thus, the identified dynamical phases might offer a new perspective: they could serve as candidate states within a larger phase space that the brain potentially traverses, possibly corresponding to different states of consciousness—an idea open to experimental testing."
>
>     3.3. (line 247) **Delete** "This perspective also informs how pathological dynamics in disorders like epilepsy or Parkinson’s might stem from altered effective connectivity, offering avenues for spectral or dynamic biomarkers."
>
>     3.4. (line 250) In the paragraph "Key limitations guide future work", we have **added** the content of "multiple phase patterns may superimpose or coexist simultaneously due to mode-coupling" as mentioned in previous rebuttal.

---

### Note · Authors · 2025-08-16

Dear Area Chair and Reviewers,

We sincerely appreciate the significant time and effort you have invested, as well as the thoughtful and constructive insights you have shared throughout the review cycle. We are grateful that the reviewers highlighted our core contributions, including:

1. Novel RMT-based Theoretical Framework for Spatially Extended Systems

    We establish a theoretical framework built upon solid foundations in Random Matrix Theory (RMT), extending its application to spatially extended neural networks.(Reviewer 1, 2, 3)

2. Biological Relevance through Minimalist Modeling

   Despite model simplicity, we show qualitative alignment with cortical dynamics observed in mice. By mapping spectral outliers to spatial modes and bulk spectrum to chaotic neural activity, our model offers a new perspective for spatial patterns of cortex neural activity. (Reviewer 1, 2)

3. Rigorous Foundational Support

   The appendices provide comprehensive analytical groundwork (e.g., utilized theorems from RMT and the derivation of the eigenspectrum; Dynamical mean-field theory implementation to overcome the limitations of linear approximation), ensuring reproducibility and offering a resource for future studies (Reviewer 2). This depth reinforces the methodological rigor underlying our numerical results.

During the rebuttal period, our detailed responses have addressed most of the reviewers' concerns. Furthermore, we humbly accept all constructive feedback and fully agree with the suggestions to highlight our "Unique sales point", avoid overstating and make clearer clarifications. Particularly, concerning quantifying similarity between the patterns obtained in simulations and experiment (Reviewer 1, 2), we have utilized order parameters including persistent homology as a quantitative measure and established a preliminary relationship between consciousness degree and the corresponding phase in our model, which will be integrated into the camera-ready version of our paper. We believe these results further enrich the significance of our theory.

We were particularly pleased to receive follow-up comments from Reviewer 1, to which we responded with our revisions point-by-point. We commit to implementing these changes and incorporating new results mentioned above into the final camera-ready version to ensure our paper is more rigorous. We are confident our revised work will be a valuable contribution to the community.

---

### Decision · Program_Chairs · 2025-09-17

**Decision:**

Accept (poster)

**Comment:**

This work attempts to characterize collective behavior in a set of spatially extended neural network models using Random Matrix Theory (RMT). The virtue of this approach is that it provides a novel approach that does not rely on homogeneous connectivity or a continuum limit. Phase diagrams are derived by considering the outliers of eigenvalue distributions of the connectivity matrices of these networks, with potential application to patterned brain states as observed in, e.g., wide field calcium imaging.

Reviewers appreciated the paper's novel theoretical contribution, which offers a complementary approach to other methods in the field and reproduces their results in overlapping cases. Weaknesses in the work were mostly presentational, with some need for contextualizing ambitious claims about consciousness and differentiating this work relative to prior art in the field.

During discussion, authors provided additional clarification of novelty and committed to revisions that should enhance the readability of the paper.